# Genome-wide mapping in a house mouse hybrid zone reveals hybrid sterility loci and Dobzhansky-Muller interactions

Leslie M Turner[1,2], Bettina Harr[1]*

[1]Department of Evolutionary Genetics, Max Planck Institute for Evolutionary Biology, Plön, Germany; [2]Laboratory of Genetics, University of Wisconsin, Madison, United States

**Abstract** Mapping hybrid defects in contact zones between incipient species can identify genomic regions contributing to reproductive isolation and reveal genetic mechanisms of speciation. The house mouse features a rare combination of sophisticated genetic tools and natural hybrid zones between subspecies. Male hybrids often show reduced fertility, a common reproductive barrier between incipient species. Laboratory crosses have identified sterility loci, but each encompasses hundreds of genes. We map genetic determinants of testis weight and testis gene expression using offspring of mice captured in a hybrid zone between *M. musculus musculus* and *M. m. domesticus*. Many generations of admixture enables high-resolution mapping of loci contributing to these sterility-related phenotypes. We identify complex interactions among sterility loci, suggesting multiple, non-independent genetic incompatibilities contribute to barriers to gene flow in the hybrid zone.

*For correspondence: harr@evolbio.mpg.de

Competing interests: The authors declare that no competing interests exist.

## Introduction

New species arise when reproductive barriers form, preventing gene flow between populations (*Coyne and Orr, 2004*). Recently, two approaches have substantially advanced the understanding of the genetic mechanisms underlying reproductive isolation (Reviewed in *Noor and Feder (2006)*; Reviewed in *Wolf et al. (2010)*). Genetic crosses in the laboratory involving model organisms have identified loci and genes causing hybrid defects, a common type of reproductive barrier caused by genetic interactions between divergent alleles at two or more loci (*Bateson, 1909*; *Dobzhansky, 1937*; *Muller, 1942*). In nature, recent technological advances enable fine-scale characterization of genome-wide patterns of divergence between incipient species and variation in hybrid zones.

For example, 'islands of divergence' have been reported in species pairs from taxonomically diverse groups (*Turner et al., 2005*; *Nadeau et al., 2011*; *Nosil et al., 2012*; *Ellegren et al., 2013*; *Hemmer-Hansen et al., 2013*; *Renaut et al., 2013*; *Carneiro et al., 2014*; *Poelstra et al., 2014*; *Schumer et al., 2014*). These high-divergence genomic outlier regions are sometimes referred to as 'islands of speciation', resistant to introgression because they harbor genes causing reproductive isolation. However, other forces can create similar genomic patterns, thus islands may not always represent targets of selection that contributed to speciation (*Noor and Bennett, 2009*; *Turner and Hahn, 2010*; *Renaut et al., 2013*; *Cruickshank and Hahn, 2014*).

An alternative approach to identify genomic regions contributing to reproductive isolation is to map known reproductive barrier traits in naturally hybridizing populations. The potential for mapping in hybrid zones is long-recognized (*Kocher and Sage, 1986*; *Harrison, 1990*; *Szymura and Barton, 1991*; *Briscoe et al., 1994*; Reviewed in *Rieseberg and Buerkle (2002)*). Hybrid zones are 'natural laboratories for evolutionary studies' (*Hewitt, 1988*), enabling investigation of speciation in progress.

**eLife digest** Different species have often evolved from a common ancestor. In order to become distinct species, however, the different groups of descendants of that ancestor must have become isolated from one another at some point in their history so that they could no longer mate or reproduce. For example, a mountain or a river might create a physical barrier that keeps species apart, so that if the species meet up again they may struggle to mate or produce offspring. Furthermore, any 'hybrid' offspring that are produced may themselves struggle to survive or successfully reproduce.

Examining the genes of the hybrid offspring that result when two recently separated species crossbreed could help us to understand how new species evolve. However, the challenges of finding enough suitable hybrids to compare means that few studies have so far investigated the genetic changes that occur to make reproduction between separate species difficult.

Two subspecies of the house mouse—*Mus musculus musculus* and *Mus musculus domesticus*—live alongside each other in a region of central Europe and can mate and produce hybrid offspring. Male hybrid mice are commonly less fertile than non-hybrids; this acts as a barrier to reproductive success that helps to maintain the separation between the two subspecies.

Turner and Harr captured wild hybrid mice, bred them in the laboratory, and studied their offspring. This strategy enabled them to measure fertility in mice very similar to wild-caught hybrids, but now all individuals can be measured at the same age and under the same environmental conditions.

A method called a genome-wide association study can be used to survey the genes of individuals with a particular disease or physical characteristic in an effort to identify gene variants that are associated with that condition. In many species, the weight of a male's testes has been linked to their fertility—small testes mean the male is likely to be less fertile. Changes in how genes are 'expressed' in the testes can also reduce fertility.

Turner and Harr used a genome-wide association study to investigate which genetic changes are linked to changes in testis weight or how genes in the testes are expressed in the offspring of hybrid mice. This revealed that many separate genetic regions are involved; including some that had not previously been identified. Turner and Harr then examined how these gene regions interact with each other. With the exception of one gene, all interacted with at least one of the other genetic regions that had been identified, forming a complex network of interactions.

Although a genome-wide association study reveals which genes are altered in hybrid mice with small testes, it does not reveal which of these genes actually cause the changes in testis size and gene expression. However, the work of Turner and Harr greatly narrows down the candidates for further investigation.

The Dobzhansky-Muller model predicts that hybrid incompatibilities between incipient species accumulate faster than linearly with time (*Orr, 1995*), thus investigating taxa in the early stages of speciation facilitates identification of incompatibilities that initially caused reproductive isolation vs incompatibilities that arose after isolation was complete.

Despite these advantages, few studies have mapped barrier traits or other fitness-related traits in nature, due to the logistical challenges of collecting dense genome-wide genetic markers in species with admixed populations and well-characterized phenotypes. Examples include associations between pollen sterility and genomic regions showing reduced introgression in a sunflower hybrid zone (*Rieseberg et al., 1999*) and loci contributing to variation in male nuptial color and body shape mapped in a recently admixed stickleback population (*Malek et al., 2012*).

House mice (*Mus musculus*) are a promising model system for genetic mapping in natural populations (*Laurie et al., 2007*) and have an abundance of genetic tools available to ultimately isolate and characterize the causative genes underlying candidate loci. Three house mouse subspecies—*M. m. musculus*, *M. m. domesticus*, and *M. m. castaneus*–diverged ~500,000 years ago from a common ancestor (Reviewed in *Boursot et al. (1993)*; *Salcedo et al. (2007)*; *Geraldes et al. (2008)*). *M. m. musculus* and *M. m. domesticus* (hereafter, *musculus* and *domesticus*) colonized Europe through different geographic routes and meet in a narrow secondary contact zone running through central Europe from Bulgaria to Denmark (*Sage et al., 1986*; *Boursot et al., 1993*). Genome-wide analyses of patterns of

gene flow in several geographically distinct transects across the hybrid zone have identified genomic regions showing reduced introgression, which may contribute to reproductive isolation (*Tucker et al., 1992*; *Macholan et al., 2007*; *Teeter et al., 2008*, *2010*; *Janousek et al., 2012*).

Reduced male fertility is common in wild-caught hybrids (*Albrechtová et al., 2012*; *Turner et al., 2012*) and in *musculus–domesticus* hybrids generated in the laboratory (*Britton-Davidian et al., 2005*; Reviewed in *Good et al. (2008a)*), implying that hybrid sterility is an important barrier to gene flow in house mice. Mapping studies using $F_1$ and $F_2$, hybrids generated from laboratory crosses between house mouse subspecies have identified many loci and genetic interactions contributing to sterility phenotypes (*Storchova et al., 2004*; *Good et al., 2008b*; *White et al., 2011*; *Dzur-Gejdosova et al., 2012*; *Turner et al., 2014*). *Prdm9*, a histone methyltransferase, was recently identified as a gene causing $F_1$ hybrid sterility and is the first mammalian hybrid incompatibility gene identified (*Mihola et al., 2009*). Comparisons between different $F_1$ crosses show that hybrid sterility alleles are polymorphic within subspecies (*Britton-Davidian et al., 2005*; *Good et al., 2008a*). Furthermore, reduced fertility phenotypes observed in nature vary in severity; complete sterility, as documented in some $F_1$ crosses, appears to be rare or absent in the hybrid zone (*Albrechtová et al., 2012*; *Turner et al., 2012*). Taken together, studies of hybrid sterility in house mice indicate that, even in the early stages of speciation, the genetic basis of hybrid defects can be complex. Studies of gene flow in the hybrid zone and of hybrid sterility in the laboratory both have advantages and have shed light on the speciation process. Mapping sterility phenotypes in natural hybrids can potentially integrate insights from the two approaches by identifying associations between hybrid incompatibility loci and reduced gene flow across the hybrid zone.

In this study, we map sterility-related phenotypes in hybrid zone mice to investigate the genetic architecture of reproductive isolation between incipient species. We performed a genome-wide association study (GWAS) to map testis weight and testis gene expression in 185 first generation lab-bred offspring of wild-caught hybrid mice (*Figure 1—figure supplement 1*). GWAS have been powerful in humans, loci contributing to hundreds of quantitative traits associated with disease and other phenotypic variation has been identified (Reviewed in *Stranger et al. (2011)*). Examples of GWAS for fitness-related traits in non-humans are only beginning to emerge (*Johnston et al., 2011*; *Filiault and Maloof, 2012*; *Magwire et al., 2012*).

Our hybrid zone GWAS identified genomic regions associated with variation in relative testis weight (testis weight/body weight) and genome-wide testis expression pattern, including regions previously implicated in hybrid sterility as well as novel loci. Motivated by the Dobzhansky–Muller genetic model of hybrid defects, we tested for genetic interactions (Dobzhansky–Muller interactions—'DMIs') between loci affecting testis weight or expression pattern. All loci except one showed evidence for interaction with at least one partner locus and most interact with more than one partner. The deviations in phenotype associated with most interactions were large–affected individuals have phenotypes below the range observed in pure subspecies–suggesting that these interactions indeed are hybrid incompatibilities. To our knowledge, this is the first GWAS for a reproductive barrier trait. Using natural hybrids provided high mapping resolution that will facilitate future studies to identify causative genes; for example, a majority of GWAS regions contain 10 or fewer genes. Moreover, this study provides the first genome-scale description of a hybrid incompatibility network in nature.

## Results

### Sterility-associated phenotypes

We investigated two phenotypes in males from the house mouse hybrid zone: relative testis weight (testis weight/body weight) and genome-wide testis gene expression pattern. Both of these phenotypes have previously been linked to hybrid male sterility in studies of offspring from crosses between *musculus* and *domesticus* and mice from the hybrid zone (*Britton-Davidian et al., 2005*; *Rottscheidt and Harr, 2007*; Reviewed in *Good et al. (2008a)*, *(2010)*; *White et al. (2011)*; *Turner et al. (2012)*, *(2014))*. We refer to these as 'sterility phenotypes', following conventional terminology in the field, however, it is important to note that the severity of defects observed in most hybrid zone mice is consistent with reduced fertility/partial sterility (*Turner et al., 2012*).

Testis expression PC1 (explaining 14.6% of the variance) is significantly correlated with relative testis weight (cor = 0.67, $P = 2 \times 10^{-16}$), indicating that there is a strong association between those two sterility phenotypes (*Figure 1—figure supplement 2*). Principal component 2 (PC2, 8.1% variance) is

strongly correlated with hybrid index (% *musculus* autosomal SNPs: cor = 0.75, $P = 2 \times 10^{-16}$), thus the effect of hybrid defects does not obscure subspecies differences in expression.

In the mapping population, 19/185 (10.2%) individuals had relative testis weight below the minimum observed in pure subspecies males and 21/179 (11.7%) individuals had expression PC1 scores below (PC1 = −46.97) the pure subspecies range.

## Association mapping

We identified four SNPs significantly associated with relative testis weight in three regions on the X chromosome using stringent thresholds determined by permutation (*Table 1*; *Figure 1A*). An additional 51 SNPs were significant using a more permissive significance threshold (false discovery rate (FDR) < 0.1). Significant SNPs were clustered in 12 genomic regions (of size 1 bp to 13.3 Mb; *Table 1*). We report GWAS regions defined using the permissive FDR threshold because we plan to combine mapping results from multiple phenotypes to identify candidate sterility loci, based on the idea that spurious associations are unlikely to be shared among phenotypes. Significant regions were located on the X chromosome and 9 autosomes, suggesting a minimum of 10 loci contribute to variation in testis weight. It is difficult to estimate the precise number of genes involved, because the extent of linkage disequilibrium (LD) of significant SNPs around a causative mutation depends on the phenotypic effect size, recombination rate, allele frequency, and local population structure. Multiple significant regions might be linked to a single causative mutation, or conversely, a significant region might be linked to multiple causative mutations in the same gene or in multiple genes.

We identified 104 SNPs on the X and chromosome 1 significantly associated with expression PC1 using stringent permutation-based thresholds (*Table 2*; *Figure 1B*). An additional 349 SNPs were significant with the more permissive threshold of FDR < 0.1. Significant SNPs clustered in 50 genomic regions located on 18 autosomes and the X.

To gain further insight into associations between sterility and gene expression, we mapped expression levels of individual transcripts. A total of 18,992/36,323 probes showed significant associations with at least one SNP. We focused on *trans* associations (SNP is located on different chromosome from transcript), based on evidence from a study in $F_2$ hybrids that *trans* expression QTL (eQTL) are associated with sterility while *cis* eQTL are predominantly associated with subspecies differences (*Turner et al., 2014*). To identify SNPs significantly enriched for *trans* associations with expression, we used a threshold set at the 95% percentile of significant probe association counts across all SNPs (i.e., SNPs that showed associations with at least 30 transcripts, *Figure 1C*).

There was substantial overlap between mapping results for testis weight and expression PC1; 48/55 SNPs significant for relative testis weight (9 regions) were also significant for expression PC1. A permutation test, performed by randomly shuffling the positions of GWAS regions in the genome, provides strong evidence that this overlap is non-random (p < 0.0001, 10,000 permutations). Most SNPs significant for testis weight and/or expression PC1 were significantly enriched for *trans* associations with individual transcripts (relative testis weight: 49/55 SNPs, 8/12 regions; PC1: 440/453 SNPs, 50/50 regions). The combined mapping results provide multiple lines of evidence for contributions of all 50 PC1 regions and 9/12 testis weight regions. The three testis-weight regions (RTW04, RTW05, RTW08) not significantly associated with testis expression phenotypes are more likely to be spurious and are weaker candidates for future study.

## Genetic interactions

Power to identify pairwise epistasis in GWAS for quantitative traits is limited even with very large sample sizes, due to multiple testing issues (e.g. *Marchini et al., 2005*). The Dobzhansky-Muller model predicts that the effect of each hybrid defect gene depends on interaction with at least one partner locus. Hence, for hybrid sterility traits, there is a hypothesis-driven framework in which to limit tests for epistasis to a small subset of possible interactions.

We tested for genetic interactions between all pairs of significant SNPs (FDR < 0.1) located on different chromosomes for testis weight and for expression PC1. We identified 142 significant pairwise interactions for relative testis weight, representing 22 pairs of GWAS regions (*Figure 2A*). These results provide evidence for a minimum of 13 autosomal–autosomal and five X–autosomal interactions affecting testis weight.

We identified 44,145 significant interactions between SNPs for expression PC1. The 913 GWAS region pairs provide evidence that at least 144 autosomal–autosomal interactions and 18 X–autosomal interactions contribute to expression PC1 (*Figure 2B*).

**Table 1.** Genomic regions significantly associated with relative testis weight

| Region*[1] | Chr | Position (Mb)† | Length (kb) | Sig. SNPs (5% perm)‡ | No. sign SNPs expression§ | Interactions# | Concordant PC1 region¶ | Concordant sterility loci** | Sterile Allele†† | No. genes (coding) ‡‡ | Candidate Genes§§ |
|---|---|---|---|---|---|---|---|---|---|---|---|
| RTW01 | 1 | 173.30–173.34 | 40.7 | 1 | 1 | 5 | PC03 | | d | 3 (3) | |
| RTW02 | 2 | 33.15 | 2.6 | 1 | 0 | 4 | PC04 | BHZ | m* | 1 (1) | |
| RTW03 | 2 | 129.59–129.65 | 59.8 | 1 | 1 | 2 | PC08 | TW[A] | d | 2 (1) | |
| RTW04 | 6 | 132.63 | – | 1 | 0 | 0 | – | | M | 0 | |
| RTW05 | 9 | 64.40 | – | 1 | 0 | 3 | – | | U | 1 (1) | |
| RTW06 | 11 | 24.25 | 0.8 | 1 | 1 | 2 | PC26 | BHZ | D | 0 | |
| RTW07 | 12 | 37.16–41.52 | 4364.2 | 4 | 4 | 7 | PC29 | | D | 20 (12) | Arl4a[EFG] |
| RTW08 | 13 | 51.44 | – | 1 | 0 | 4 | – | TW[B] | d | 0 | |
| RTW09 | 17 | 56.68–58.44 | 1752.2 | 4 | 2 | 8 | PC43 | SCbin[A]; TW[A]; BHZ | M | 42 (39) | Acsbg2[E]; Clpp[G]; Safb[G]; Tmem146[EG] |
| RTW10 | X | 12.17 | – | 1 (1) | 1 | 4 | PC46 | ASH[D]; eQTLHS[C]; HT[A]; SC[A] | m* | 1 (1) | |
| RTW11 | X | 85.13–98.43 | 13294.3 | 35 (2) | 35 | 3 | PC49 | ASH[D]; DBT[A]; eQTLHS[C]; FERT[B]; HT[A]; PBT[A]; SC[B]; TAS[A]; TW[BD]; BHZ | D | 191 (67) | A[EFG]; Arx[G]; Pcyt1b[EFG]; Tex11[EFG]; Zfx[EFG] |
| RTW12 | X | 127.57–134.13 | 6555.5 | 4 (1) | 4 | 2 | PC50 | ASH[D]; eQTLHS[C]; shPC1[A]; SC[D]; TW[D]; BHZ | D | 158 (71) | Nxf2[G]; Taf7l[EFG] |

*Significant SNPs <10 Mb apart were combined into regions.

†Significant intervals were defined by positions of the most proximal and distal SNPs with LD > 0.9 to a significant SNP.

‡The number of SNPs significant at FDR < 0.1 is reported; number of significant SNPs significant with <0.05 P value in permutations is in parentheses.

§Number of significant SNPs enriched for associations with transcripts expressed on another chromosome (P < 0.05; FDR < 0.1; >30 transcripts).

#Number of regions with significant interactions.

¶(Overlapping regions significant for expression PC1 (see **Table 2**).

**Sterility QTL overlapping or within 10 Mb from [A](**White et al. 2011**), [B](**Dzur-Gejdosova et al. 2011**), [C](**Turner et al. 2014**), [D](**Good et al. 2008b**). Abbreviations for phenotypes: ASH: abnormal sperm head morphology, TW: testis weight, SC: sperm count, shPC1: sperm head shape PC1, eQTLHS: *trans* eQTL hotspot, FERT: fertility, PBT: proximal bent sperm tail, HT: headless/tailless sperm, DBT: distal bent sperm tail, TAS: total abnormal sperm. BHZ: overlapping candidate regions with evidence from epistasis in the Bavarian hybrid zone transect (**Janousek et al. 2012**).

††Sterile allele inferred on the basis of frequency of a majority of significant SNPs in pure subspecies samples: D–domesticus; M–musculus; lower-case indicates $F_{ST}$< 0.7 between pure subspecies; * indicates overlapping PC1 region is D sterile; U–nondiagnostic SNP and/or no majority allele.

‡‡Number of genes (protein-coding) overlapping region.

§§Genes with roles in male reproduction on the basis of [E]male reproduction gene ontology terms (see 'Materials and methods') or phenotypes of knockout models reported in [F](**Matzuk and Lamb 2008**) or [G]MGI database.

**Source data 1.** Protein-coding genes in significant relative testis weight regions.

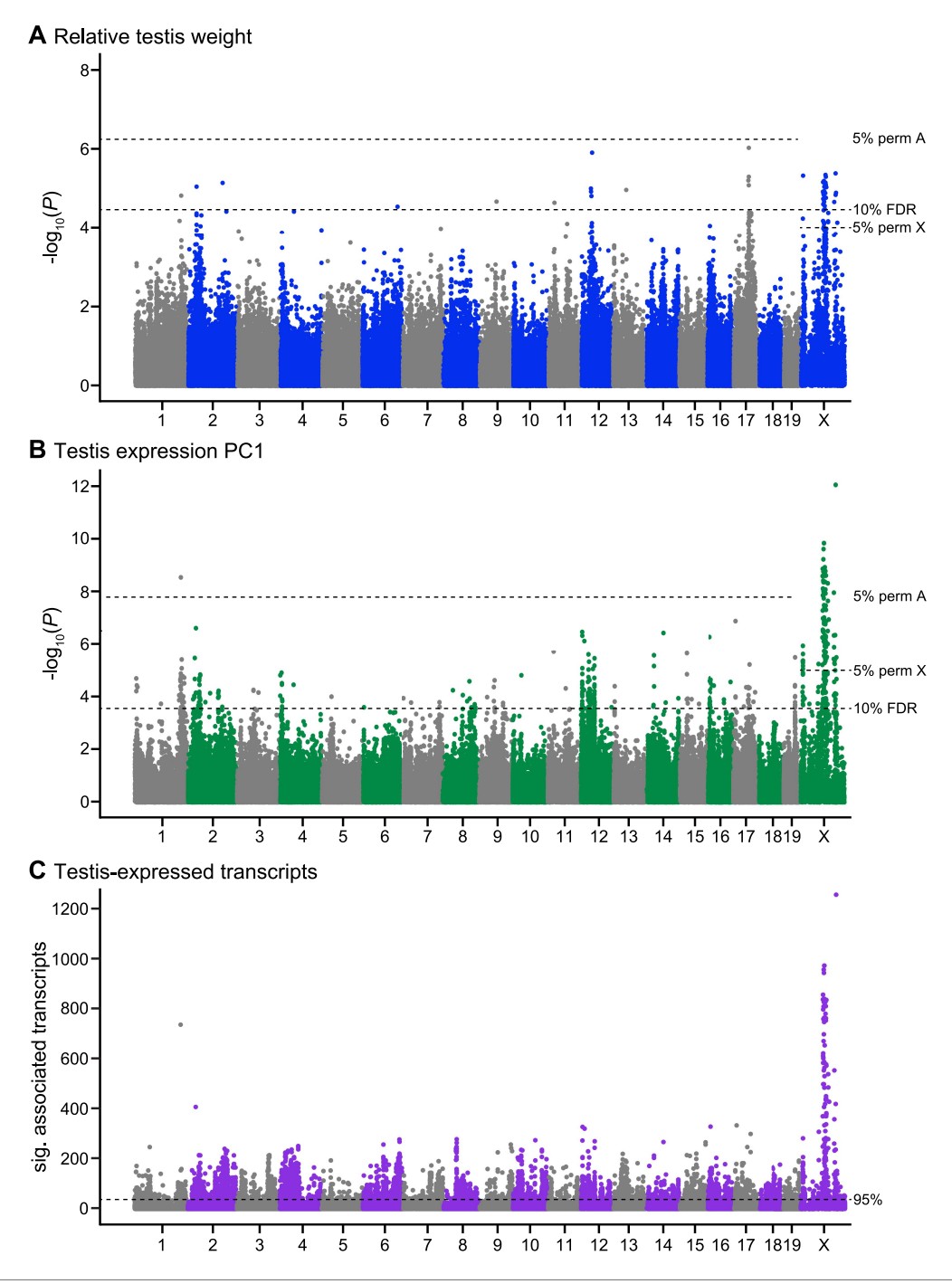

**Figure 1**. Manhattan plot of GWAS results. Single SNPs associated with (**A**) relative testis weight, (**B**) testis expression principal component 1, and (**C**) expression of transcripts located on other chromosomes (*trans*). Dashed lines indicate significance thresholds based on: permutations for autosomes (labeled 5% perm A), permutations for X chromosome (labeled 5% perm X), false discovery rate <0.1 (labeled 10% FDR), and 95th percentile of significant transcript association counts across SNPs (labeled 95%).

The following source data and figure supplements are available for figure 1:

**Source data 1**. SNPs significantly associated with relative testis weight and/or testis expression PC1 (excel file).

*Figure 1. Continued on next page*

*Figure 1. Continued*

**Figure supplement 1**. Geographic location of and genetic makeup of mapping population.

**Figure supplement 2**. Principal components analysis of genome-wide gene expression in testis.

## Effect size

We used deviations from population means for single SNPs and two-locus genotypes to estimate the phenotypic effects of GWAS regions and interactions (*Figure 3A,B*).

As expected, interactions had greater effects, on average, than single loci for both phenotypes (relative testis weight: single locus mean = −1.81 mg/g, interaction mean = −4.07 mg/g; expression PC1: single locus mean = −81.51, interaction mean = −130.77). We provide examples of autosomal–autosomal and X–autosomal SNP pairs with significant interactions for each phenotype in *Figure 3C*. It is important to note that mean deviations are rough estimates of effect sizes, which don't account for family structure.

It is possible that some of the GWAS regions we mapped contribute to quantitative variation within/between subspecies, rather than hybrid defects. The lowest genotypic means for most interactions fell below the range observed in pure subspecies (relative testis weight: 19/22 (86.3%) region pairs; expression PC1: 877/913 (96%) region pairs; *Figure 3A,B*), consistent with the hypothesis that interactions represent Dobzhansky–Muller incompatibilities.

## Mapping simulations

We performed simulations to assess the performance of the mapping procedure for different genetic architectures by estimating the power to detect causative loci and the false positive rate (*Figure 4*). We simulated phenotypes based on two-locus genotypes from the SNP dataset using genetic models for nine genetic architecture classes (i.e. autosomal vs X linked, varied dominance) with parameters based on the observed distribution of relative testis weight (*Figure 4—figure supplement 1*, *Figure 4—source data 1*).

The distribution of distances to the causal SNP for all significant SNPs located on the same chromosome (*Figure 4—figure supplement 2*) shows that the majority of significant SNPs (62.7%) are within 10 Mb of the causal SNP, however, a small proportion of significant SNPs are >50 Mb from the causal SNP. In most cases, causal SNPs detected at long distances also had significant SNPs nearby, for example 83.4% of loci with significant SNPs 1–10 Mb distant also have significant SNPs within 1 Mb. These results provide support for our choice to define significant GWAS regions by combining significant SNPs within 10 Mb, and suggest that these regions are likely to encompass the causative gene.

As expected, the power to detect one or both causative loci depended on the location (autosomal vs X-linked), dominance, and frequency of both 'causative' alleles (*Figure 4*, *Table 3*). For example, the mean percentage of simulations for which both loci were detected (SNP < 10 Mb significant by permutation-based threshold) was six times higher (14.4%) for the X chromosome × autosomal-dominant architecture compared to the autosomal-recessive × autosomal-recessive architecture (2.6%). The relationship between power and the proportion of affected individuals in the mapping population was complex. Interestingly, power was high for some simulations with very few affected individuals. In these cases, the few individuals carrying the lower frequency sterility allele by chance also carried the sterility allele from the second locus, thus the average single allele effects were not diminished by individuals carrying one but not both interacting sterility alleles.

It is important to note that our empirical results suggest that the two-locus models used to simulate phenotypes are overly simplified. We predict that involvement of a sterility locus in multiple incompatibilities would reduce the influence of allele/genotype frequencies of any single partner locus on power.

To estimate the false discovery rate from simulations, we classified significant SNPs not located on the same chromosome as one of the causative SNPs as false positives. Choosing an appropriate distance threshold for false vs true positives on the same chromosome was not obvious given the distribution of distances to causal SNPs (*Figure 4*-figure supplement 2). We classified significant SNPs <50 Mb from causative SNPs as true positives and excluded SNPs >50 Mb when calculating FDR. Using permutation-based significance thresholds, the median false positive rate was

**Table 2.** Genomic regions significantly associated with testis expression PC1

| Region* | Chr | Position (Mb)† | Length (kb)‡ | Sig. SNPs (5% perm)‡ | No. sign SNPs expression§ | Interactions# | Concordant RTW region¶ | Concordant sterility loci** | Sterile Allele†† | No. genes (coding) ‡‡ | Candidate Genes§§ |
|---|---|---|---|---|---|---|---|---|---|---|---|
| PC01 | 1 | 8.01–12.72 | 4715.2 | 4 | 4 | 46 | | BHZ | U | 40 (18) | Mybl1$^{GH}$ |
| PC02 | 1 | 99.53 | – | 1 | 1 | 19 | | BHZ | D | 0 | |
| PC03 | 1 | 166.84–185.83 | 18988.2 | 28 (1) | 25 | 47 | RTW01 | BHZ | D | 297 (229) | Adcy10$^{FGH}$; Atp1a4$^{H}$; Ddr2G$^{H}$; DeddH; Exo1$^{FGH}$; F11r$^{G}$; H3f3a$^{GH}$; Lbr$^{H}$; Lmx1a$^{H}$; Mael$^{FH}$; Mpz$^{H}$; Vangl2$^{H}$ |
| PC04 | 2 | 21.72–49.01 | 27288.0 | 30 | 30 | 45 | RTW02 | TW$^{A}$; BHZ | D | 604 (334) | Acvr2a$^{GH}$; Bmyc$^{H}$; Grin1$^{H}$; Il1rn$^{i}$; Lhx3$^{H}$; Notch1$^{F}$; Nr5a1$^{GH}$; Nr6a1$^{F}$; Odf2$^{FH}$; Pax8$^{GH}$; Sh2d3c$^{H}$; Sohlh1$^{GH}$; Strbp$^{FGH}$; Tsc1$^{H}$ |
| PC05 | 2 | 67.00 | – | 1 | 1 | 38 | | TW$^{A}$ | d | 1 (0) | |
| PC06 | 2 | 84.56–84.68 | 125.6 | 1 | 1 | 38 | | eQTLHS$^{C}$; TW$^{A}$ | d | 8 (7) | |
| PC07 | 2 | 114.21–116.79 | 2579.4 | 7 | 7 | 41 | | eQTLHS$^{C}$; TW$^{A}$; BHZ | D | 20 (4) | |
| PC08 | 2 | 129.59–129.65 | 59.8 | 1 | 1 | 16 | RTW03 | TW$^{A}$ | d | 2 (1) | |
| PC09 | 3 | 63.61–63.62 | 5.5 | 2 | 2 | 36 | | DBT$^{A}$ | M | 1 (1) | |
| PC10 | 3 | 82.14 | – | 1 | 1 | 22 | | eQTLHS$^{C}$ | d | 0 | |
| PC11 | 4 | 3.14–11.16 | 8023.3 | 8 | 8 | 44 | | | D | 98 (31) | Ccne2$^{H}$; Chd7$^{H}$; Plag1$^{H}$ |
| PC12 | 4 | 52.80 | – | 1 | 1 | 33 | | | U | 0 | |
| PC13 | 5 | 37.81 | – | 1 | 1 | 40 | | | m | 1 (1) | |
| PC14 | 6 | 5.78–5.90 | 121.6 | 1 | 1 | 28 | | BHZ | d | 1 (1) | |
| PC15 | 7 | 7.09–7.10 | 9.0 | 1 | 1 | 24 | | shPC1$^{A}$ | d | 1 (1) | |
| PC16 | 7 | 35.47 | – | 1 | 1 | 21 | | shPC1$^{A}$ | d | 1 (1) | |
| PC17 | 7 | 140.36–140.98 | 620.9 | 3 | 3 | 43 | | | D | 8 (7) | |
| PC18 | 8 | 37.56 | – | 1 | 1 | 22 | | STA$^{A}$ | d | 1 (1) | |
| PC19 | 8 | 74.15–74.17 | 20.9 | 1 | 1 | 33 | | STA$^{A}$ | d | 1 (1) | |
| PC20 | 8 | 90.23–106.77 | 16539.6 | 5 | 3 | 45 | | STA$^{A}$; BHZ | D | 146 (101) | Bbs2$^{GH}$; Ccdc135$^{F}$; Csnk2a2$^{GH}$; Katnb1$^{H}$; Nkd1$^{FH}$ |
| PC21 | 8 | 118.11–120.56 | 2451.0 | 2 | 2 | 40 | | STA$^{A}$ | U | 23 (16) | |
| PC22 | 9 | 32.44 | – | 1 | 1 | 34 | | BHZ | m | 0 | |
| PC23 | 9 | 57.23–60.59 | 3359.6 | 5 | 4 | 41 | | | D | 69 (54) | 2410076l21Rik$^{F}$; Bbs4$^{GH}$; Cyp11a1$^{GH}$ |
| PC24 | 9 | 91.04–91.22 | 180.0 | 2 | 2 | 33 | | | D | 0 | |
| PC25 | 10 | 34.9–35.08 | 185.2 | 1 | 1 | 27 | | PBT$^{A}$ | d | 0 | |
| PC26 | 11 | 24.25 | 0.8 | 1 | 1 | 29 | RTW06 | BHZ | D | 0 | |

*Table 2. Continued on next page*

*Table 2. Continued*

| Region* | Chr | Position (Mb)† | Length (kb) | Sig. SNPs (5% perm)‡ | No. sign SNPs expression§ | Interactions# | Concordant RTW region¶ | Concordant sterility loci** | Sterile Allele†† | No. genes (coding) ‡‡ | Candidate Genes§§ |
|---|---|---|---|---|---|---|---|---|---|---|---|
| PC27 | 11 | 67.99–69.47 | 1479.7 | 1 | 1 | 31 | | shPC1^A | D | 67 (46) | Aurkb^H, Odf4^F, Shbg^F, Trp53^H |
| PC28 | 12 | 7.85–16.13 | 8278.4 | 19 | 19 | 47 | | | D | 54 (32) | Apob^FGH, Gdf7^GH, Pum2^H |
| PC29 | 12 | 28.99–54.22 | 25238.3 | 46 | 44 | 47 | RTW07 | BHZ | D | 150 (93) | Ahr^GH, Arl4a^GH, Immp2l^FGH, Slc26a4^H |
| PC30 | 12 | 116.53 | – | 1 | 1 | 35 | | | m | 0 | |
| PC31 | 13 | 6.74–6.85 | 113.3 | 2 | 2 | 35 | | TW^A | D | 0 | |
| PC32 | 14 | 29.53–32.21 | 2675.5 | 5 | 4 | 43 | | STA^A, TW^B | D | 44 (35) | Chdh^H; Dnahc1^G, Tkt^H |
| PC33 | 14 | 66.74–75.01 | 8274.9 | 2 | 2 | 41 | | SC^B | U | 98 (71) | Fndc3a^FGH, Gnrh1^GH, Npm2^F, Piwil2^FGH, Rb1^H |
| PC34 | 14 | 121.69–121.77 | 83.2 | 1 | 1 | 31 | | | d | 1 (1) | |
| PC35 | 15 | 27.75–31.46 | 3701.3 | 5 | 5 | 47 | | HT^A, TAS^A | D | 19 (8) | |
| PC36 | 15 | 45.67 | – | 1 | 1 | 36 | | HT^A, TAS^A | d | 0 | |
| PC37 | 15 | 73.00 | – | 1 | 1 | 27 | | eQTLHS^C | d | 1 (1) | |
| PC38 | 16 | 8.18–18.51 | 10329.1 | 56 | 56 | 41 | | BHZ | D | 201 (132) | Prm1^FGH, Prm2^FGH, Prm3^F, Ranbp1^H, Rimbp3^H, Rpl39^F, Snai2^H, Spag6^FGH, Tnp2^FGH, Top3b^I, Tssk1^FH, Tssk2^FH |
| PC39 | 16 | 29.16–29.17 | 9.7 | 1 | 1 | 34 | | | d | 0 | |
| PC40 | 16 | 66.52–66.53 | 13.1 | 2 | 2 | 39 | | STA^A | U | 0 | |
| PC41 | 16 | 90.92–90.93 | 11.6 | 1 | 1 | 35 | | STA^A | d | 1 (1) | |
| PC42 | 17 | 11.05–11.18 | 132.7 | 3 | 3 | 43 | | eQTLHS^C, FERT^B, SC^AB, TW^AB | Dh | 1 (1) | |
| PC43 | 17 | 42.08–63.29 | 21217.1 | 13 | 11 | 45 | RTW09 | SC^A, TW^A, BHZ | Md | 272 (209) | Acsbg2^F, Clpp^H, Dazl^FGH, Klhdc3^F, Mea1^F, Pot1b^H, Safb^H, Sgol1^F, Tcte1^H, Tdrd6^H, Tmem146^F, Ubr2^FGH, Zfp318^H |
| PC44 | 17 | 77.34–83.59 | 6248.8 | 2 | 2 | 33 | | TW^A | D | 53 (36) | |
| PC45 | 19 | 44.82–45.74 | 918.1 | 10 | 9 | 46 | | BHZ | D | 23 (16) | Btrc^GH, Dpcd^H |
| PC46 | X | 11.34–19.34 | 7995.3 | 19 (7) | 19 | 44 | RTW10 | ASH^D, eQTLHS^C, HT^A, SC^AD, TW^D, BHZ | D | 82 (21) | |
| PC47 | X | 36.94 | – | 1 | 1 | 28 | | ASH^D, eQTLHS^C, FERT^B, HT^A, shPC1^A, SC^ABD, TW^BD, BHZ | d | 0 | |

*Table 2. Continued on next page*

Table 2. Continued

| Region* | Chr | Position (Mb)† | Length (kb) | Sig. SNPs (5% perm)‡ | No. sign SNPs expression§ | Interactions# | Concordant RTW region¶ | Concordant sterility loci** | Sterile Allele†† | No. genes (coding) ‡‡ | Candidate Genes§§ |
|---|---|---|---|---|---|---|---|---|---|---|---|
| PC48 | X | 68.03–70.77 | 2742.2 | 4 (1) | 3 | 43 | | ASH[AE]; DBT[A]; eQTLHS[C]; FERT[B]; HT[A]; OFF[E]; PBT[A]; SC[BE]; TAS[A]; TW[BE]; BHZ | U | 62 (41) | Cetn2[F]; Mtm1[H] |
| PC49 | X | 83.62–108.53 | 24911.7 | 125 (84) | 125 | 45 | RTW11 | DBT[A]; eQTLHS[C]; FERT[B]; HT[A]; PBT[A]; shPC1[A]; SC[B]; TAS[A]; TW[BD]; BHZ | D | 407 (142) | Ar[GH]; Arx[H]; Atp7a[H]; Pcyt1b[FGH]; Tex11[FGH]; Tsx[.H]; Zfx[FGH] |
| PC50 | X | 127.01–137.37 | 10365.1 | 21 (11) | 21 | 45 | RTW12 | ASH[D]; eQTLHS[C]; shPC1[A]; SC[D]; TW[D]; BHZ | D | 212 (92) | Nxf2[H]; Taf7l[FGH]; Tsc22d3[H] |

*Significant SNPs <10 Mb apart were combined into regions.

†Significant intervals were defined by positions of the most proximal and distal SNPs with LD > 0.9 to a significant SNP.

‡The number of SNPs significant at FDR < 0.1 is reported; number of significant SNPs significant with <0.05 P value in permutations is in parentheses.

§Number of significant SNPs enriched for associations with transcripts expressed on another chromosome (P < 0.05; FDR < 0.1; >30 transcripts).

#Number of regions with significant interactions.

¶Overlapping regions significant for relative testis weight (see **Table 1**).

**Sterility QTL overlapping or within 10 Mb from [A](**White et al. 2011**), [B](**Dzur-Gejdosova et al. 2011**), [C](**Turner et al. 2014**), [D](**Good et al. 2008b**), [E](**Storchova et al. 2004**). Abbreviations for phenotypes: ASH: abnormal sperm head morphology, TW: testis weight, SC: sperm count, shPC1: sperm head shape PC1, eQTLHS: trans eQTL hotspot, STA: seminiferous tubule area, FERT: fertility, PBT: proximal bent sperm tail, HT: headless/tailless sperm, DBT: distal bent sperm tail, TAS: total abnormal sperm, OFF: number of offspring. BHZ: overlapping candidate regions with evidence from epistasis in the Bavarian hybrid zone transect (**Janousek et al. 2012**).

††Sterile allele inferred on the basis of frequency of a majority of significant SNPs in pure subspecies samples: D–domesticus; M–musculus; lower-case indicates FST < 0.7 between pure subspecies; * indicates overlapping PC1 region is D sterile; U–nondiagnostic SNP and/or no majority allele; Dh–two SNPs with domesticus sterile alleles, one SNP heterozygous genotype shows sterile pattern; Md–majority musculus sterile alleles but some SNPs diagnostic domesticus sterile alleles.

‡‡Number of genes (protein-coding) overlapping region.

§§Genes with roles in male reproduction on the basis of [F]male reproduction gene ontology terms (see 'Materials and methods') or phenotypes of knockout models reported in [G](**Matzuk and Lamb 2008**) or [H]MGI database.

**Source data 1**. Protein-coding genes in significant testis expression PC1 regions.

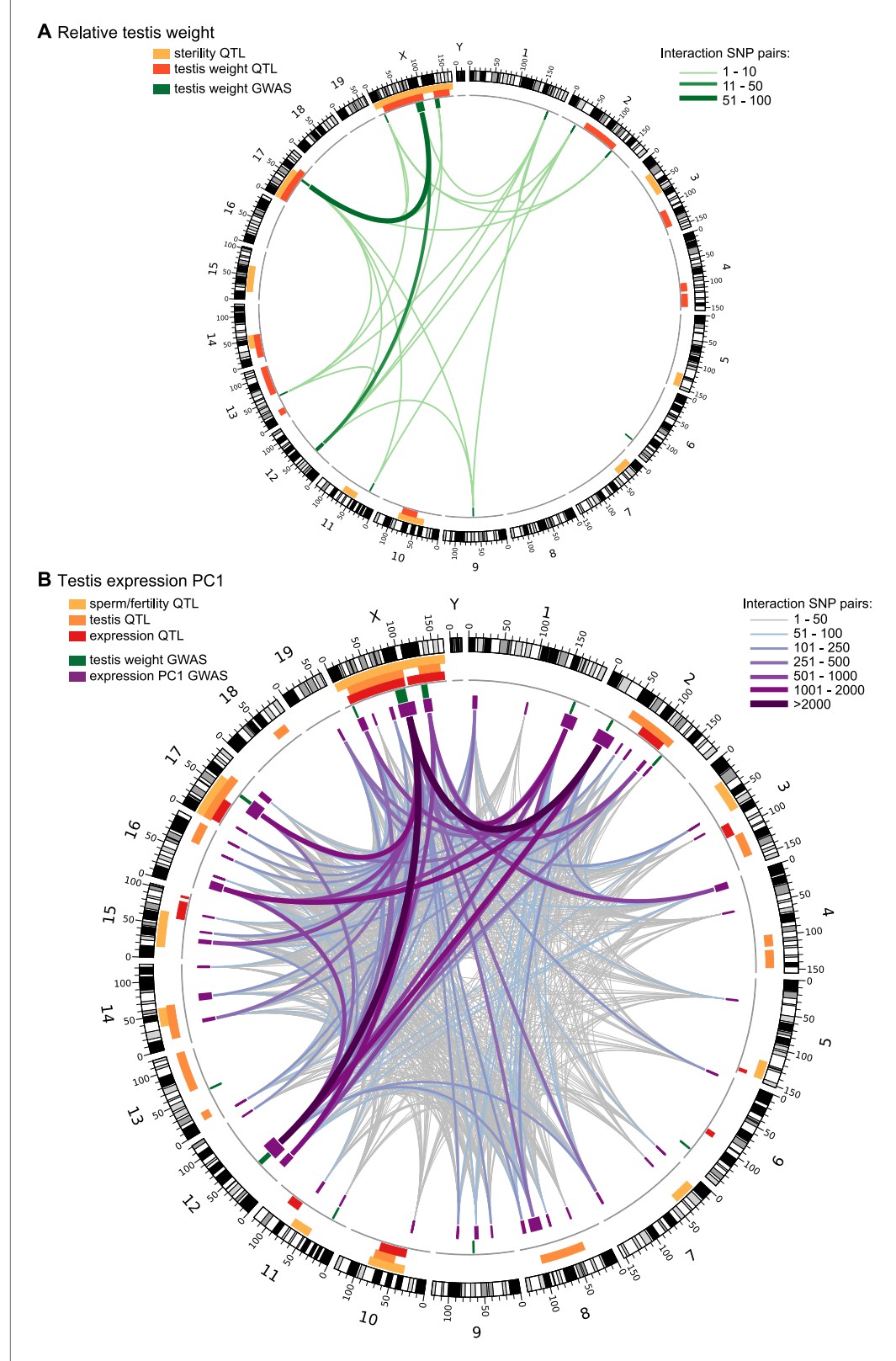

**Figure 2**. Significant GWAS regions and interactions in hybrid zone mice. Results for (**A**) relative testis weight and (**B**) testis expression principal component 1 in hybrid zone mice. In (**A**) orange and yellow boxes in outer rings (outside grey line) indicate quantitative trait loci (QTL) identified for testis weight and other sterility phenotypes in *Figure 2. Continued on next page*

*Figure 2. Continued*

previous studies (see **Table 1** for details). Green boxes indicate significant GWAS regions for relative testis weight. Green lines represent significant genetic interactions between regions; shade and line weight indicate the number of significant pairwise interactions between SNPs for each region pair. In (**B**) orange boxes in outer rings indicate QTL for testis-related phenotypes (testis weight and seminiferous tubule area) identified in previous studies, yellow boxes indicate QTL for other sterility phenotypes and red boxes indicate *trans* eQTL hotspots (see **Table 2** for details). Green boxes indicate significant GWAS regions for relative testis weight. Purple boxes indicate significant GWAS regions for testis expression PC1. Lines represent significant genetic interactions between regions; color and line weight—as specified in legend—indicate the number of significant pairwise interactions between SNPs for each region pair. Plot generated using circos (**Krzywinski et al., 2009**).

The following source data and figure supplements are available for figure 2:

**Source data 1**. Significant genetic interactions (SNP pairs) for relative testis weight (excel file).

**Source data 2**. Significant genetic interactions (SNP pairs) for testis expression PC1 (excel file).

**Figure supplement 1**. Genetic interactions associated with hybrid sterility in hybrid zone mice and in $F_2$ hybrids.

0.014 (calculated for simulations with ≥10 SNPs within 50 Mb of either causative locus). These results suggest that significant SNPs from the GWAS identified using this more stringent threshold are likely to be true positives. By contrast, the median false positive rate was 0.280 using the FDR < 0.1 threshold, indicating this threshold is more permissive than predicted. Thus, there is a substantial chance that SNP associations with relative testis weight and expression PC1 identified using this threshold are spurious and evidence is weak for GWAS regions comprising one SNP significantly associated with a single phenotype.

## Discussion

Genetic mapping of testis weight and testis gene expression in hybrid zone mice implicated multiple autosomal and X-linked loci and a complex set of interactions between loci. These results provide insight into the genetic architecture of a reproductive barrier between two incipient species in nature.

### Association mapping in natural hybrid populations

The potential to leverage recombination events from generations of intercrossing in hybrid zones to achieve high-resolution genetic mapping of quantitative traits has been recognized for decades (Reviewed in **Rieseberg and Buerkle, (2002)**). Until recently, collection of dense genotype datasets and large sample sizes has not been feasible in natural populations due to logistics and costs. This study demonstrates that loci and genetic interactions contributing to reproductive barrier traits can be identified in a GWAS with a modest sample size (see also related study mapping craniofacial phenotypes in this mapping population **Pallares et al. 2014**). Sample sizes approximating those used for human GWAS are not necessary if the prevalence and genetic architecture of the trait of interest are favorable. In general, epistasis makes genetic mapping more difficult. However, for hybrid defects, dependence of the phenotype on epistasis conversely may facilitate mapping. Despite substantial deleterious effects in hybrids, incompatibility alleles are not subject to negative selection within species and may be at high frequency or fixed within species. Hence, the prevalence of affected individuals in a hybrid zone for epistatic traits may be much higher than for deleterious traits in pure populations (e.g. disease in humans).

Combining mapping of multiple sterility-related phenotypes substantially improved power to identify sterility loci. We identified a few loci for each phenotype using stringent significance thresholds based on permutation. In addition, most loci identified using more permissive thresholds showed significant associations with more than one phenotype. Spurious associations are unlikely to be shared across phenotypes, thus evidence from multiple phenotypes provided confidence for contributions of nine genomic regions to testis weight (on the X and 5 autosomes) and 50 genomic regions to expression PC1 (on the X and 18 autosomes).

The high resolution of mapping in the hybrid zone provides an advantage over laboratory crosses. For example, significant regions identified here (median = 2.1 Mb, regions with defined intervals) are

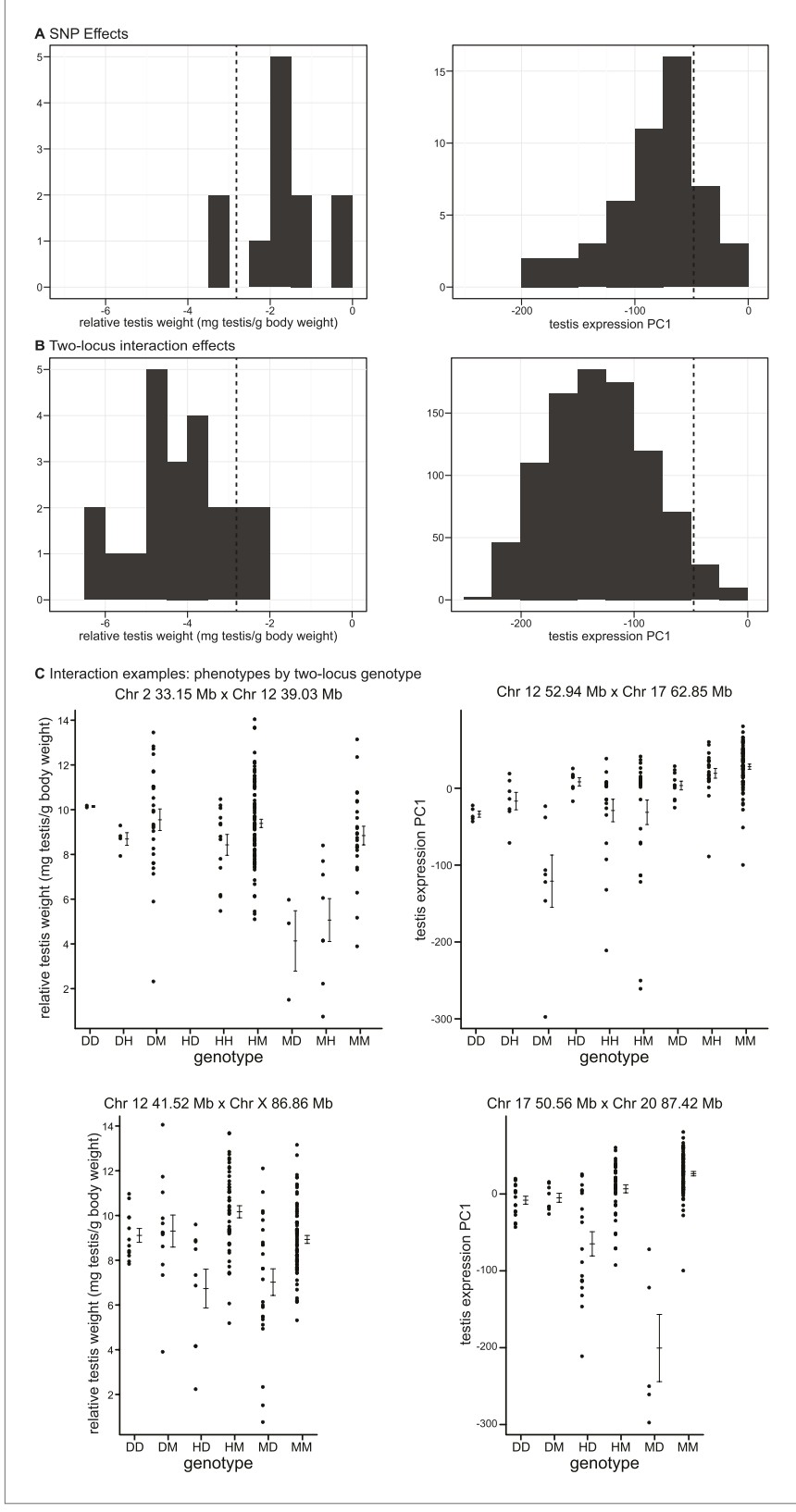

**Figure 3**. Phenotypic effects of testis-weight loci and interactions. Histograms showing maximum deviations from the population mean for (**A**) single SNPs and (**B**) two-locus interactions. Dashed vertical lines indicate minimum

*Figure 3. Continued*

values observed in pure subspecies males. (**C**) Examples of phenotypic means by two-locus genotype for autosomal–autosomal and X–autosomal interactions. Genotypes are indicated by one letter for each locus: D—homozygous for the *domesticus* allele, H—heterozygous, M—homozygous *musculus*.

much smaller than sterility QTL identified in $F_2$s (35.1 Mb; *White et al., 2011*). Many GWAS regions contain few enough genes that it will be possible to individually evaluate the potential role of each in future studies to identify causative genes. For example, 8/12 testis-weight regions and 28/50 expression PC1 regions contain 10 or fewer protein-coding genes.

We identified candidate genes with known roles in reproduction in four testis-weight regions and 17 expression PC1 regions (*Tables 1–2*). However, for the majority of regions (8/12 relative testis weight, 33/50 expression PC1), there are no overlapping/nearby genes previously linked to fertility. It is unlikely that these regions would be prioritized if contained in large QTL intervals. High resolution mapping is possible using mapping resources such as the collaborative cross (*Aylor et al., 2011*) and heterogeneous stocks (*Svenson et al., 2012*), but these populations represent a small proportion of genetic diversity in house mice (*Yang et al., 2011*) and hybrid incompatibility alleles may have been lost during strain production.

## Polymorphism of hybrid male sterility loci

Comparisons of different $F_1$ crosses between strains of *domesticus* and *musculus* have shown that hybrid sterility phenotypes and loci depend on the geographic origins of parental strains (*Britton-Davidian et al., 2005*; *Good et al., 2008a*), suggesting that most hybrid sterility alleles are segregating as polymorphisms within subspecies. Several of the loci identified in this study of hybrid zone mice are novel, providing additional evidence that sterility alleles are polymorphic within subspecies. However, a majority of loci we identified in natural hybrids are concordant with previously identified sterility QTL (*Tables 1–2*, *Figure 2*). This similarity suggests that there are common genetic factors underlying hybrid sterility in house mice, although there was no statistical support that genome-wide patterns of overlap with previous studies for testis weight or expression PC1 were non-random ($p > 0.05$, 10,000 permutations).

*Prdm9*, discovered by mapping $F_1$ hybrid sterility, is the only characterized hybrid sterility gene in mice (*Mihola et al., 2009*). None of the GWAS regions identified here overlap *Prdm9* (chromosome 17, 15.7 Mb). However, one expression PC1 region (PC42) is ~4 Mb proximal to *Prdm9*. Reductions in PC1 are observed in individuals that are heterozygous or homozygous for the *domesticus* allele at PC42. This pattern is partially consistent with sterility caused by *Prdm9*, which occurs in heterozygous individuals carrying sterile alleles from *domesticus* (*Dzur-Gejdosova et al., 2012*; *Flachs et al., 2012*). We did not find evidence for significant associations between SNPs near *Prdm9* and testis weight; the nearest GWAS region (RTW09) is ~41 Mb distal and low testis-weight is associated with the *musculus* allele.

There is concordance between some of the genetic interactions between loci identified here and interactions identified by mapping sterility phenotypes and testis expression traits in an $F_2$ cross between *musculus* and *domesticus* (*White et al., 2011*; *Turner et al., 2014*) (*Figure 2—figure supplement 1*). Precise overlap between some GWAS regions and interaction regions from $F_2$s identifies strong candidates for future studies to identify the causative mechanisms and genes underlying sterility loci. For example, an interaction between chromosome 12 and the central X chromosome (RTW11, PC49) identified for testis weight and expression PC1 overlaps an interaction affecting testis expression in $F_2$ hybrids (*Turner et al., 2014*). The 4.3 Mb interval of overlap among chromosome 12 loci (RTW07, PC29, 32.38–41.43 Mb $F_2$s) encompasses 12 protein-coding genes, including a gene with a knockout model showing low testis weight and sperm count (*Arl4a*) (*Schurmann et al., 2002*), and two genes with roles in regulating gene expression (*Meox2, Etv1*).

We compared the positions of GWAS regions to 182 regions (163 autosomal, 19 X-linked) with evidence for epistasis based on a genome-wide analysis of genomic clines in a transect across the house mouse hybrid zone in Bavaria (*Janousek et al., 2012*), the same region where the progenitors of the mapping population were collected. Five testis-weight regions and 18 expression-PC1 regions overlap candidate regions from the hybrid zone genomic clines analysis (*Tables 1–2*), however, the patterns of overlap were not statistically significant ($p > 0.05$, 10,000 permutations).

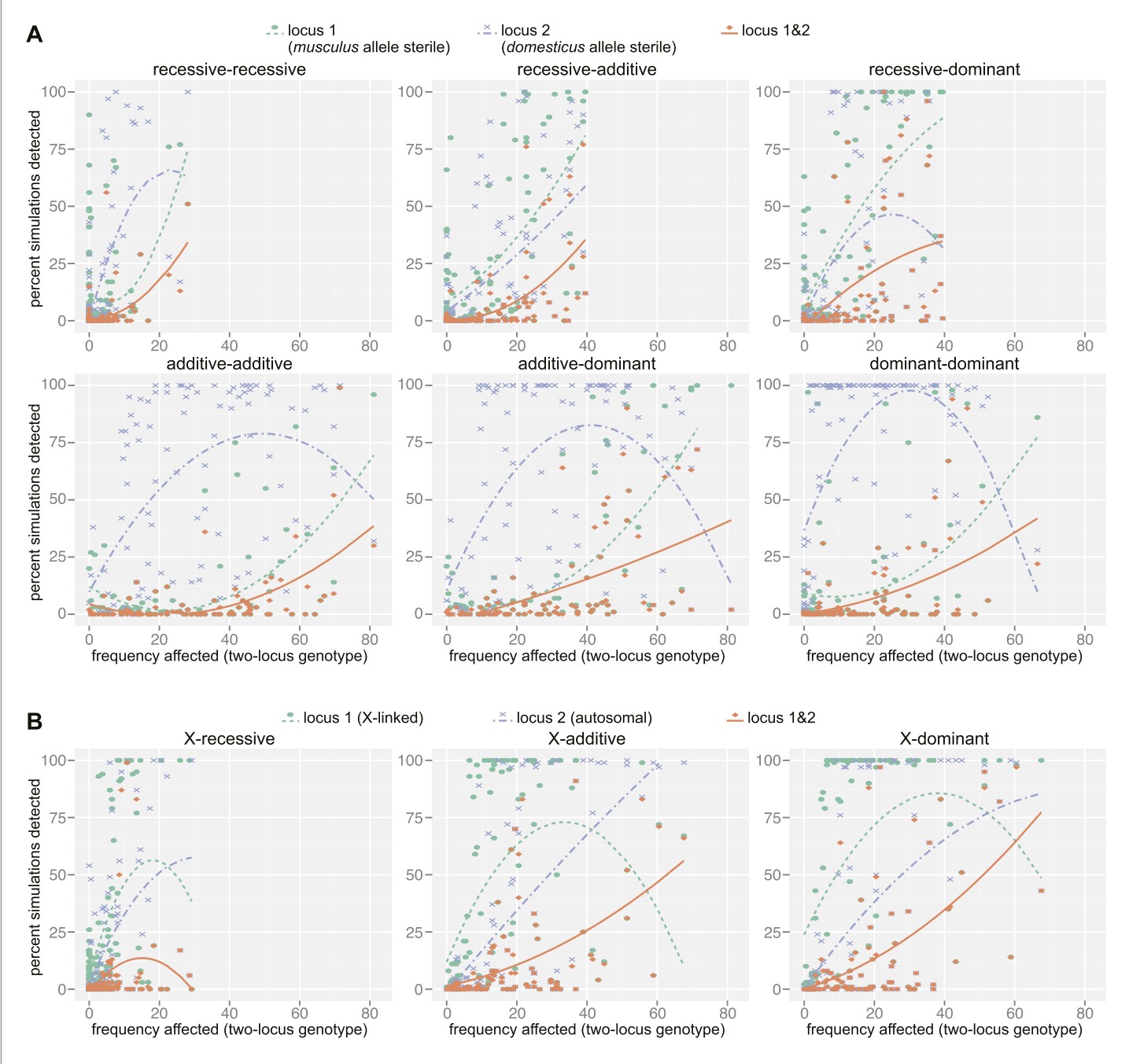

**Figure 4**. Mapping power in simulations. Each panel illustrates results from a single genetic architecture model for (**A**) 100 autosomal–autosomal SNP pairs and (**B**) 100 X—autosomal SNP pairs. Each point represents the percentage of data sets generated from a single SNP pair in which locus 1 (*domesticus* sterile allele; green), locus 2 (*musculus* sterile allele; purple), or both loci (orange) were identified by association mapping (≥1 SNP significant by permutation based threshold within 10 Mb of 'causal' SNP). The x axis indicates the percentage of individuals with partial or full sterility phenotypes. Curves were fit using second order polynomials. In (**A**), locus 1 indicates the SNPs with *musculus* alleles sterile and locus 2 indicates the SNPs with *domesticus* alleles sterile. In (**B**), locus 1 is the X-linked SNP and locus 2 is the autosomal SNP.

The following source data and figure supplements are available for figure 4:

**Source data 1**. Z scores for simulation models.

**Figure supplement 1**. Mapping simulation methods.

*Figure 4. Continued on next page*

*Figure 4. Continued*

**Figure supplement 2**. Distances of significant SNPs to causal SNP in simulations.

Future introgression analyses using high-density markers within and around GWAS regions may be useful in identifying causative genes and estimating the contributions of sterility alleles to reduced gene flow.

## Role of the X chromosome

Three GWAS regions associated with testis weight and five expression PC1 regions are located on the X chromosome. The X-chromosomal regions surpass the stringent permutation-based significance thresholds and thus have strong statistical support. These results are consistent with evidence for an important role for the X in hybrid sterility from laboratory crosses between subspecies strains geographically diverse in origin (*Guenet et al., 1990*; *Elliott et al., 2001*; *Oka et al., 2004*, *2007*; *Storchova et al., 2004*; *Good et al., 2008a*, 2008b; *Mihola et al., 2009*; *White et al., 2012*) and evidence for greatly reduced gene flow of X-linked loci across the European hybrid zone (*Tucker et al., 1992*; *Payseur et al., 2004*; *Macholan et al., 2007*; *Teeter et al., 2008*, *2010*). A disproportionately

**Table 3.** Results of mapping simulations

| Architecture* | Med. Distance to Causal SNP (Mb) X chromosome/ Autosome | Locus 1 detected†,‡ | | | Locus 2 detected‡,§ | | | Both loci detected‡ | | | Mean No. Sig. SNPs | | |
|---|---|---|---|---|---|---|---|---|---|---|---|---|---|
| | | 0.2 Mb | 1 Mb | 10 Mb | 0.2 Mb | 1 Mb | 10 Mb | 0.2 Mb | 1 Mb | 10 Mb | 10 Mb# | 50 Mb# | Diff. Chr.¶ |
| Permutation P<0.05 | | | | | | | | | | | | | |
| rec-rec | 5.9 | 7.2 | 8.4 | 12.3 | 9.0 | 11.7 | 15.8 | 0.3 | 0.7 | 2.6 | 1.1 | 1.5 | 5.5 |
| rec-add | 2.6 | 18.3 | 22.2 | 28.0 | 12.6 | 15.8 | 21.0 | 3.2 | 4.4 | 7.2 | 3.5 | 2.3 | 4.4 |
| rec-dom | 2.0 | 27.4 | 31.8 | 39.2 | 19.1 | 22.2 | 26.4 | 5.5 | 7.8 | 12.9 | 6.9 | 8.5 | 5.0 |
| add-add | 1.4 | 6.7 | 7.7 | 10.5 | 47.5 | 51.9 | 55.8 | 2.7 | 3.1 | 4.7 | 7.9 | 9.2 | 6.1 |
| add-dom | 1.7 | 14.2 | 15.9 | 19.0 | 51.6 | 55.7 | 59.2 | 6.0 | 7.5 | 10.3 | 11.1 | 13.3 | 5.4 |
| dom-dom | 1.8 | 7.8 | 9.8 | 14.3 | 63.8 | 66.9 | 70.6 | 2.4 | 3.7 | 7.3 | 14.7 | 17.6 | 6.2 |
| X-rec | 12.2/4.8 | 10.3 | 14.0 | 26.2 | 10.0 | 12.7 | 18.8 | 0.1 | 1.3 | 4.9 | 5.6 | 9.9 | 4.8 |
| X-add | 9.1/2.0 | 33.9 | 39.1 | 48.5 | 24.3 | 25.6 | 31.0 | 3.8 | 5.3 | 11.4 | 21.9 | 35.7 | 5.7 |
| X-dom | 9.8/2.0 | 46.5 | 51.3 | 59.7 | 26.9 | 28.5 | 32.6 | 5.9 | 8.6 | 14.4 | 31.0 | 52.8 | 3.8 |
| FDR <0.1 | | | | | | | | | | | | | |
| rec-rec | 10.0 | 16.6 | 21.4 | 34.7 | 18.5 | 23.5 | 35.5 | 3.5 | 5.4 | 15.0 | 5.1 | 8.3 | 34.7 |
| rec-add | 5.5 | 32.7 | 39.7 | 52.7 | 27.2 | 32.6 | 45.2 | 11.4 | 15.5 | 27.9 | 13.2 | 18.9 | 32.9 |
| rec-dom | 4.1 | 42.2 | 49.7 | 62.9 | 33.5 | 37.2 | 48.4 | 16.5 | 21.3 | 33.8 | 22.2 | 30.1 | 28.7 |
| add-add | 3.6 | 14.4 | 17.6 | 30.6 | 63.3 | 69.3 | 77.6 | 8.4 | 11.3 | 23.3 | 21.6 | 28.8 | 36.8 |
| add-dom | 3.5 | 26.5 | 31.1 | 42.0 | 65.5 | 70.6 | 78.1 | 18.2 | 22.8 | 33.5 | 29.2 | 39.3 | 29.1 |
| dom-dom | 3.6 | 16.4 | 22.1 | 35.3 | 76.8 | 79.8 | 85.9 | 9.4 | 15.1 | 29.3 | 35.5 | 48.0 | 26.5 |
| X-rec | 12.2/7.8 | 10.3 | 14.0 | 26.2 | 20.0 | 25.2 | 40.5 | 0.7 | 3.1 | 11.0 | 10.3 | 17.5 | 34.6 |
| X-add | 9.1/4.7 | 33.9 | 39.1 | 48.5 | 33.2 | 36.6 | 48.3 | 6.3 | 9.4 | 20.9 | 28.7 | 46.1 | 30.0 |
| X-dom | 9.8/5.0 | 46.5 | 51.3 | 59.7 | 37.0 | 41.2 | 50.9 | 11.4 | 16.3 | 27.2 | 38.8 | 65.5 | 21.6 |

*Architecture abbreviations: add–additive; dom–dominant; rec–recessive.
†Locus 1 for autosomal pairs is *musculus* sterile allele; locus 1 for X-autosomal pairs is X-linked.
‡3'detected'–≥1 significant SNP within given distance criterion.
§Locus 2 for autosomal pairs has a *domesticus* sterile allele; locus 2 for X-autosomal pairs is autosomal.
#Mean number significant SNPs within distance criterion for either locus.
¶Mean number significant SNPs on chromosomes not containing 'causal' SNPs.

large contribution of the X chromosome is a common feature of reproductive isolation in many taxa, the so-called 'large X effect' (*Coyne and Orr, 1989*).

The *musculus* derived X chromosome has been implicated repeatedly in genetic studies of sterility in $F_1$ and $F_2$ hybrids (Reviewed in *Good et al. (2008a)*; *White et al. (2011)*). By contrast, *domesticus* alleles were associated with the sterile pattern for most loci we identified on the X in hybrid zone mice (*Tables 1–2*). A testis expression-QTL mapping study performed in $F_2$s also showed that *domesticus* ancestry in the central/distal region of the X was associated with a sterile expression pattern (*Turner et al., 2014*). Differences between studies might reflect geographic variation in sterility alleles, but identification of *domesticus*-sterile X alleles only in generations beyond the $F_1$ suggests that interactions with recessive autosomal partner loci are essential. The importance of recessive sterility alleles was demonstrated previously by the discovery of multiple novel recessive loci in an $F_2$ mapping study (*White et al., 2011*). $F_1$ hybrids are essentially absent in nature (*Teeter et al., 2008*; *Turner et al., 2012*) because the hybrid zone is ≥30 km wide (*Boursot et al., 1993*), thus pure subspecies individuals rarely encounter each other. Consequently, recessive autosomal loci acting in $F_2$ and advanced generation hybrids contribute to the maintenance of reproductive isolation in the hybrid zone and may have played important roles in its establishment.

## Genetic architecture of hybrid sterility

Despite a growing list of sterility loci and genes identified in a variety of animal and plant taxa, there are few cases of Dobzhansky–Muller incompatibilities for which all partner loci are known (*Phadnis, 2011*). Hence, there remain many unanswered questions about the genetic architecture of hybrid defects. For example, how many incompatibilities contribute to reproductive barriers in the early stages of speciation? How many partner loci are involved in incompatibilities? Are these patterns consistent among taxa?

The interactions contributing to sterility phenotypes we mapped in hybrid zone mice reveal several general features of the genetic architecture of hybrid sterility. Most sterility loci interact with more than one partner locus. This pattern is consistent with evidence from studies mapping sterility in $F_1$ *musculus–domesticus* hybrids (*Dzur-Gejdosova et al., 2012*) and mapping interactions affecting testis gene expression in $F_2$ hybrids (*Turner et al., 2014*). We did not have sufficient power to map interactions requiring three or more sterility alleles, but interactions between alleles from the same subspecies imply their existence. Loci causing male sterility in *Drosophila pseudoobscura* Bogota–USA hybrids also have multiple interaction partners; seven loci of varying effect size interact to cause sterility (*Phadnis, 2011*). In hybrids between *Drosophila koepferae* and *Drosophila buzzatii*, sterility is associated with many loci of small effect, consistent with a polygenic threshold model (*Moran and Fontdevila, 2014*).These studies suggest that biological pathways/networks are often affected by multiple Dobzhansky–Muller interactions; a single pairwise interaction between incompatible alleles disrupts pathway function enough to cause a hybrid defect phenotype, but when more incompatible alleles are present the effects of multiple pairwise interactions are synergistic. Variation in the effect sizes of sterility loci might then reflect variation in the number of networks in which the gene is involved and the connectedness/centrality of the gene within those networks.

Characteristics of the incompatibility network are important for generating accurate models of the evolution of reproductive isolation. A 'snowball effect'—faster-than-linear accumulation of incompatibilities caused by epistasis—is predicted on the basis of the Dobzhansky–Muller model (*Orr, 1995*; *Orr and Turelli, 2001*). Patterns of accumulation of hybrid incompatibilities in *Drosophila* and *Solanum* provide empirical support for the snowball hypothesis (*Matute et al., 2010*; *Moyle and Nakazato, 2010*). Because most GWAS regions have many interaction partners, our results are not consistent with the assumption of the snowball model that incompatibilities are independent, suggesting that network models of incompatibilities (*Johnson and Porter, 2000*; *Porter and Johnson, 2002*; *Johnson and Porter, 2007*; *Palmer and Feldman, 2009*) may be more accurate for understanding the evolution of reproductive barriers in house mice.

Involvement of hybrid sterility loci in interactions with multiple partner loci also has important implications for understanding the maintenance of the hybrid zone. Because deleterious effects of a sterility allele are not dependent on a single partner allele, the marginal effect of each locus and thus visibility to selection are less sensitive to the allele frequencies at any single partner locus in the population.

Identifying and functionally characterizing incompatibility genes is an important goal in understanding speciation but is unrealistic in most non-model organisms. By contrast, mapping reproductive

isolation traits in natural populations to identify the number and location of loci and interactions is feasible. General features of the genetic architecture of hybrid sterility—the number of incompatibilities and number and effect size of interacting loci—are arguably more likely to be shared among organisms than specific hybrid sterility genes. Comparison of these features among taxa may reveal commonalities of the speciation process.

## Materials and methods

### Mapping population

The mapping population includes first-generation lab-bred male offspring of mice captured in the hybrid zone (Bavaria) in 2008 (*Turner et al., 2012*) (*Figure 1—figure supplement 1*). We included 185 mice generated from 63 mating pairs involving 37 unrelated females and 35 unrelated males. Many dams and sires were used in multiple mating pairs, thus our mapping population includes full siblings, half siblings, and unrelated individuals. Most mating pairs (53 pairs, 149 offspring) were set up with parents originating from the same or nearby trapping locations. Eleven pairs (36 offspring) include dams and sires originating from more distant trapping locations; phenotypes of these offspring were not reported in *Turner et al. (2012)*.

### Phenotyping

Males were housed individually after weaning (28 days) to prevent effects of dominance interactions on fertility. We measured combined testis weight and body weight immediately after mice were sacrificed at 9–12 weeks. We calculated relative testis weight (testis weight/body weight) to account for a significant association between testis weight and body weight (Pearson's correlation = 0.29, p = $4.9 \times 10^{-5}$).

We classify individuals with relative testis weight below the range observed in pure subspecies as showing evidence for sterility (*Turner et al., 2012*). To confirm that this is an appropriate threshold for inferring hybrid defects, we compared this value to relative testis weights reported previously for offspring from intraspecific and interspecific crosses (*Good et al., 2008a*). The pure subspecies minimum we observed is substantially lower (>2 standard deviations) than means for males from intraspecific crosses (converted from single relative testis weight: *musculus*[PWK] × *musculus*[CZECH] − mean = 10.2, standard deviation = 1.2; *domesticus*[LEWES] x *domesticus*[WSB] − mean = 11.0, standard deviation = 1.0) and comparable to (within 1 standard deviation) values observed in $F_1$ hybrids from 4/7 interspecific crosses that showed significant reductions (mean plus one standard deviation 4.6–9.2 mg/g).

### Testis gene expression

We measured gene expression in testes of 179 out of the 185 males from the mapping population. Freshly dissected testes were stored in RNAlater (Qiagen, Hilden, Germany) at 4°C overnight, then transferred to −20°C until processed. We extracted RNA from 15–20 mg whole testis using Qiagen RNeasy kits and a Qiagen Tissue Lyser for the homogenization step. We verified quality of RNA samples (RIN >8) using RNA 6000 Nano kits (Agilent) on a 2100 Bioanalyzer (Agilent, Waldbronn, Germany).

We used Whole Mouse Genome Microarrays (Agilent) to measure genome-wide expression. This array contains 43,379 probes surveying 22,210 transcripts from 21,326 genes. We labeled, amplified, and hybridized samples to arrays using single-color Quick-Amp Labeling Kits (Agilent), according to manufacturer protocols. We verified the yield (>2 µg) and specific activity (>9.0 pmol Cy3/µg cRNA) of labeling reactions using a NanoDrop ND-1000 UV-VIS Spectrophotometer (NanoDrop, Wilimington, DE, USA). We scanned arrays using a High Resolution Microarray Scanner (Agilent) and processed raw images using Feature Extraction Software (Agilent). Quality control procedures for arrays included visual inspection of raw images and the distribution of non-uniformity outliers to identify large spatial artifacts (e.g. caused by buffer leakage or dust particles) and quality control metrics from Feature Extraction protocol GE1_QCMT_Dec08.

We mapped the 41,174 non-control probe sequences from the Whole Mouse Genome Microarray to the mouse reference genome (NCBI37, downloaded March 2011) using BLAT ((*Kent, 2002*); minScore = 55, default settings for all other options). Probes with multiple perfect matches, more than nine imperfect matches, matches to non-coding/intergenic regions only, or matches to more than one gene were excluded. A total of 36,323 probes (covering 19,742 Entrez Genes) were retained.

We preformed preprocessing of microarray data using the R package Agi4x44PreProcess (*Lopez-Romero, 2009*). We used the background signal computed in Feature Extraction, which incorporates

a local background measurement and a spatial de-trending surface value. We used the 'half' setting in Agi4x44PreProcess, which sets intensities below 0.5 to 0.5 following background subtraction and adds an offset value of 50. Flags from Feature Extraction were used to filter probes during preprocessing (wellaboveBG = TRUE, isfound = TRUE, wellaboveNEG = TRUE). We retained probes with signal above background for at least 10% of samples. We used quantile normalization to normalize signal between arrays. Expression data were deposited in Gene Expression Omnibus as project GSE61417.

To identify major axes of variation in testis expression, we performed a principal components analysis using *prcomp* in R (*R Development Core Team 2010*) with scaled variables.

## Genotyping

We extracted DNA from liver, spleen, or ear samples using salt extraction or DNeasy kits (Qiagen). Males from the mapping population were genotyped using Mouse Diversity Genotyping Arrays (Affymetrix, Santa Clara, CA) by Atlas Biolabs (Berlin, Germany).

We called genotypes at 584,729 SNPs using *apt-probeset-genotype* (Affymetrix) and standard settings. We used the *MouseDivGeno* algorithm to identify variable intensity oligonucleotides (VINOs) (*Yang et al., 2011*); 53,148 VINOs were removed from the dataset. In addition, we removed 18,120 SNPs with heterozygosity >0.9 in any population because these SNPs likely represent additional VINOs. We performed additional filtering steps on SNPs included in the dataset used for mapping. We only included SNPs with a minor allele frequency >5% in the mapping population. SNPs without a genome position or with missing data for >15% of the individuals in the mapping population or pure subspecies reference panel were removed. We pruned the dataset based on linkage disequilibrium (LD) to reduce the number of tests performed. LD pruning was performed in PLINK (*Purcell et al., 2007*; Purcell n.d.) using a sliding window approach (30 SNPs window size, 5 SNPs step size) and a VIF threshold of $1 \times 10^{-6}$ (VIF = $1/(1-R^2)$, where $R^2$ is the multiple correlation coefficient for a SNP regressed on all other SNPs simultaneously). This procedure essentially removed SNPs in perfect LD. These filtering steps yielded 156,204 SNPs.

## Ancestry inference

To identify ancestry-informative SNPs, we compared genotypes from 21 pure *M. m. domesticus* individuals (11 from Massif Central, France and 10 from Cologne/Bonn, Germany) and 22 *M. m. musculus* individuals (11 from Námest nad Oslavou, Czech Republic and 11 from Almaty, Kazakhstan) (*Staubach et al., 2012*).

We used *Structure* (*Pritchard et al., 2000*; *Falush et al., 2003*) to graphically represent the genetic composition of our mapping population (*Figure 1—figure supplement 1*). We included one diagnostic SNP per 20 cM, 3–5 markers/chromosome totaling 60 SNPs genome wide. We used the 'admix' model in *Structure* and assumed two ancestral populations.

## Association mapping

To identify genomic regions significantly associated with relative testis weight and testis gene expression, we used a mixed model approach to test for single SNP associations. Admixture mapping—often applied in studies using samples with genetic ancestry from two distinct populations—was not appropriate for this study because it was not possible to account for relatedness among individuals in the mapping population (*Buerkle and Lexer, 2008*; *Winkler et al., 2010*).

We performed association mapping using GEMMA (*Zhou and Stephens, 2012*), which fits a univariate mixed model, incorporating an *n* x *n* relatedness (identity-by-state) matrix as a random effect to correct for genetic structure in the mapping population. We estimated relatedness among the individuals in the mapping population in GEMMA using all markers and the –gk 1 option, which generates a centered relatedness matrix. For each single SNP association test we recorded the Wald test p value. Phenotypes tested include relative testis weight (testis weight/body weight, RTW), testis expression principal component 1 (PC1, 14.6% variance, associated with fertility, *Figure 1—figure supplement 2*), and normal quantile ranks of gene expression values for individual transcripts. Neither RTW nor expression PC1 was significantly correlated with age at phenotyping (RTW: cor = −0.02, p = 0.72; PC1: cor = 0.01, p = 0.90), thus we did not include age in the model. SNP data, phenotypic data, and kinship matrix to run GEMMA area are available through Dryad at: doi:10.5061/dryad.2br40.

To account for multiple testing, we first determined stringent significance thresholds by permutation. We randomized phenotypes among individuals 10,000 times, recording the lowest p value on the X and the lowest p value on any autosome for each permutation. Thresholds set to the fifth

percentile across permutations for RTW were $5.73 \times 10^{-7}$ (autosomes) and $5.83 \times 10^{-5}$ (X chromosome); thresholds for expression PC1 were $1.66 \times 10^{-8}$ (autosomes) and $1.01 \times 10^{-5}$ (X chromosome). Next, we identified regions using a more permissive significance threshold based on the 10% false discovery rate (*Benjamini and Hochberg, 1995*), equivalent to $p = 3.49 \times 10^{-5}$ for RTW and $p = 2.86 \times 10^{-4}$ for expression PC1.

To estimate the genomic interval represented by each significant LD-filtered SNP, we report significant regions defined by the most distant flanking SNPs in the full dataset showing $r^2 > 0.9$ (genotypic LD, measured in PLINK) with each significant SNP. We combined significant regions <10 Mb apart into a single region.

## Testing for genetic interactions

Identifying genetic interactions using GWAS is computationally and statistically challenging. To improve power, we reduced the number of tests performed by testing for interactions only among significant SNPs (FDR < 0.1) identified using GEMMA. We tested all pairs of significant SNPs located on different chromosomes for each phenotype (692 pairs RTW, 82,428 pairs expression PC1). To account for relatedness among individuals we used a mixed model approach, similar to the model implemented in GEMMA. We used the *lmekin* function from the *coxme* R package (*Therneau, 2012*) to fit linear mixed models including the identity-by-state kinship matrix as a random covariate. We report interactions as significant for SNP pairs with $p < 0.05$ and FDR < 0.1 for interaction terms (RTW: FDR < 0.1 ~ $p < 0.02$; expression PC1: FDR < 0.09 ~ $p < 0.05$).

## Mapping simulations

We performed simulations to evaluate the performance of our mapping approach under varying genetic architectures and allele frequencies. We simulated phenotypes using several genetic models of two-locus epistasis and parameters based on the empirical distribution of relative testis weight. The simulation procedure is illustrated in *Figure 4—figure supplement 1*. To preserve genetic structure, we simulated phenotypes using two-locus genotypes from the SNP dataset.

We tested 100 autosomal–autosomal SNP pairs (SNPs on different chromosomes) and 100 X–autosomal pairs (50 with *domesticus* X-linked sterile alleles and 50 with *musculus* X-linked sterile alleles). The criteria used for choosing 'causative' SNPs were a minor allele frequency >0.05 in the mapping population and fixed in at least one subspecies. The 'sterile' allele could be polymorphic or fixed within subspecies but the alternate 'non-sterile' allele had to be fixed within the other subspecies—e.g. *domesticus* sterile alleles have frequencies 0.05–1.0 in the *domesticus* reference populations from France and Germany and the alternate allele at those SNPs are fixed in *musculus* samples from the Czech Republic and Kazakhstan. For each pair, the 'causative' SNPs were randomly selected from all SNPs meeting those criteria (144,506 possible *domesticus* sterile, 124,390 possible *musculus* sterile).

For each SNP pair, we modeled all possible combinations of recessive, additive, and dominant sterile alleles. For each model type, we assigned mean Z scores for each possible two-locus genotype (*Figure 4—source data 1*). The magnitude of the most severe phenotype (−2.3 standard deviations) is based on observed relative testis weights in the most severely affected males. The mean Z score for heterozygotes in additive models was −1.15. Mean Z scores for non-sterile genotypes in the models were randomly drawn from a uniform distribution between −0.5 and 0.5.

For each SNP pair/architecture, 100 data sets were generated by drawing phenotypes (Z scores) for each individual from a normal distribution with the appropriate two-locus mean and standard deviation = 0.75. The standard deviation value, equivalent to 2.98 mg/g, was chosen on the basis of standard deviations in pure subspecies samples from the mapping population (*domesticus* = 2.13, *musculus* = 3.65; (*Turner et al., 2012*)). This value is higher than standard deviations in intraspecific $F_1$ males (*domesticus*LEWES × *domesticus*WSB = 1.2, *musculus*PWK × *musculus*CZECH = 1.0; (*Good et al., 2008a*)), suggesting estimates of mapping power may be conservative.

In total, 90,000 simulations were performed, (9 architectures × 100 SNP pairs × 100 data sets). We identified significant SNPs for each data set using GEMMA, as described above for the empirical data.

## Significance of overlap between candidate sterility loci

We used permutations to test for non-random co-localization of candidate sterility loci from this study and previous QTL and hybrid zone studies. The locations of significant GWAS regions for relative testis

weight and expression PC1 were randomized 10,000 times using BEDTools (*Quinlan and Hall, 2010*). To assess overlap between significant regions for the two phenotypes, we counted the number of RTW regions overlapping PC1 regions (and vice versa) for each permutation. To test for overlap between GWAS identified regions and previously reported candidate regions for related phenotypes, we counted the number of permuted regions overlapping the positions of the published regions (fixed) for each replicate. GWAS regions for both phenotypes were compared to genomic regions with evidence for epistasis and reduced introgression in the Bavarian transect of the hybrid zone (*Janousek et al., 2012*). In addition, RTW regions were compared to testis weight QTL from mapping studies in $F_2$ hybrids from crosses between subspecies (*Storchova et al., 2004*; *Good et al., 2008b*; *White et al., 2011*; *Dzur-Gejdosova et al., 2012*) and expression PC1 regions were compared to *trans* eQTL hotspots identified in $F_2$ hybrids (*Turner et al., 2014*).

## Gene annotation

We used ENSEMBL (version 66, February 2012) Biomart to download gene annotations for genomic regions significantly associated with relative testis weight. We identified candidate genes in significant regions with roles in male reproduction using reviews of male fertility (*Matzuk and Lamb, 2008*), manual searches, MouseMine searches for terms related to male fertility (http://www.mousemine.org/), and gene ontology (GO) terms related to male reproduction or gene regulation (plus children): meiosis GO:0007126; DNA methylation GO:0006306; regulation of gene expression GO:0010468; transcription GO:0006351; spermatogenesis GO:0007283; male gamete generation GO:0048232; gamete generation GO:0007276; meiotic cell cycle GO:0051321. Many genes with roles in reproduction reported in publications were not annotated with related GO terms, highlighting the limitations of gene ontology. Moreover, genes causing sterility might not have functions obviously related to reproduction.

## Acknowledgements

We thank Xiang Zhou for expert support using GEMMA. We thank Bret Payseur for useful discussion and Diethard Tautz, Luisa Pallares, Trevor Price, Detlef Weigel, Jiri Forejt, Gil McVean and an anonymous reviewer for comments on the manuscript. LMT was supported by postdoctoral funding from the Max Planck Society (to D Tautz) and by a National Human Genome Research Institute (NHGRI) training grant in Genomic Sciences to the University of Wisconsin (NHGRI 5T32HG002760). Research funding for this project was provided by the Max Planck Society (to D Tautz) and the Deutsche Forschungsgemeinschaft (SFB-680 to BH).

SNP genotype data were deposited in doi:10.5061/dryad.2br40 and gene expression data in Gene Expression Omnibus GSE61417.

## Additional information

### Funding

| Funder | Grant reference number | Author |
|---|---|---|
| Deutsche Forschungsgemeinschaft | SFB-680 | Bettina Harr |
| Max-Planck-Gesellschaft | | Leslie M Turner, Bettina Harr |
| National Human Genome Research Institute | 5T32HG002760 | Leslie M Turner |

The funders had no role in study design, data collection and interpretation, or the decision to submit the work for publication.

### Author contributions

LMT, Conception and design, Acquisition of data, Analysis and interpretation of data, Drafting or revising the article; BH, Conception and design, Analysis and interpretation of data, Drafting or revising the article

### Ethics

Animal experimentation: This study was performed according to approved animal protocols and institutional guidelines of the Max Planck Institute. Mice were maintained and handled in accordance

to FELASA guidelines and German animal welfare law (Tierschutzgesetz § 11, permit from Veterinäramt Kreis Plön: 1401-144/PLÖ-004697).

## Additional files

### Major datasets

The following dataset was generated

| Author(s) | Year | Dataset title | Dataset ID and/or URL | Database, license, and accessibility information |
|---|---|---|---|---|
| Turner L and Harr B | 2014 | GWAS mapping genotype data | doi:10.5061/dryad.2br40 | Available at Dryad Digital Repository under a CC0 Public Domain Dedication |
| Turner L and Harr B | 2014 | Testis expression in *Mus musculus domesticus* x *Mus musculus musculus* hybrid zone mice | GSE61417; http://www.ncbi.nlm.nih.gov/geo/query/acc.cgi?acc=GSE61417 | Publicly available at GEO (http://www.ncbi.nlm.nih.gov/geo/) |

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
