## [Decision Letter]

Thank you for sending your work entitled “Genome-wide mapping in a natural hybrid zone reveals hybrid male sterility loci and Dobzhansky-Muller interactions” for consideration at *eLife*. Your article has been favorably evaluated by Detlef Weigel (Senior editor) and 3 reviewers, one of whom is a member of our Board of Reviewing Editors.

The Reviewing editor and the other reviewers discussed their comments before we reached a decision. We all feel that the design and direction of your experiment are important innovations and the work has the potential to be important in the field. However, there are a number of substantive points that need to be addressed. Moreover, the reviewers all felt that there were multiple aspects of analysis, presentation and interpretation that need to be improved.

Below, you will find the full reviews. I should note that it is not standard practice at *eLife* to return the full reviews in addition to the editor's summary. However, in this case I think the range of comments is a useful reflection of the responses of readers and are likely to be helpful in revision.

In terms of revising, the following major points need to be addressed:

Analysis: Given the large effect of the X chromosome it seems critical to include X variants in the covariance matrix. We also believe that more could be made of analyses of the genetic architecture of the trait (e.g. contribution of individual chromosomes to variance in the trait using GCTA or similar software). Similarly the DMI model makes specific predictions about the direction of epistatic effects (combinations of derived alleles deleterious) which should be easy to address by polarising variants using an outgroup.

Interpretation: The phenotype studied is not hybrid sterility (despite the Title). There is actually no direct evidence for an association between the trait studied and fitness. Indeed the statement that there were no significant results for sperm count is a little worrying. However, there is limited evidence that the phenotypes of the hybrids are typically outside the range of normal variation within the species. It is important to note these caveats.

Presentation: Without fine-mapping or replication, the ability to identify/localise specific genes as being important in the trait is limited. For this reason, we feel the emphasis on long lists of rather weakly-supported candidate genes is misplaced. There are also issues with formatting, clarity of figures and citations. Please concentrate on these above points in your revision. The comments below should help you in understanding details of these issues.

Reviewer #1:

The genetic study of reproductive isolation and speciation has the potential to be transformed by the large-scale study of genetic variation in hybrid zones and mapping experiments. This paper presents the first (at least as far as I know) GWAS of a reproductive isolation phenotype (relative testis weight) between Mus musculus musculus and M. m. domesticus from a hybrid zone. The principle findings are:

1) There are multiple (estimate of 26) regions across the genome contributing to the phenotype.

2) Some of the identified regions overlap with (or are close to) regions identified as affecting reproductive traits from F2 crosses.

3) Many of the regions harbour loci that also affect gene expression in the testis.

4) There is some evidence for interactions between loci, consistent with the DMI model.

These findings are of broad interest to the field and an interesting counterpart to the mapping results, though perhaps not unexpected and I feel not presented in as compelling a way as possible (see below). To be compelling, I think there has to be a more rigorous approach to validating the findings.

Major points:

1) A standard GWAS approach, using linear mixed models to account for relatedness between individuals is used. However, while human GWAS studies typically use 'genome-wide significant' (a somewhat vague term, but implying a conservative threshold for significance of 5e-8) and require replication, the approach taken here is to use an FDR approach with a generous threshold of 10% and no replication. Permutation studies presented show that this threshold is somewhat permissive. More generally, there is no attempt made to either establish how well calibrated the FDR approach is, nor to validate findings through replication. To me this is a serious limitation, and while I understand the authors' desire to maximise the findings from the study, I think a more limited, but well-validated, set of loci would be both more credible and of wider value. In short, I think validation of association signals in a replication set of samples is needed.

2) I would like to know more about the data. For example, how does relative testis weight vary with age (mice were killed across a range of ages), does it differ between the parental species? Likewise, more could be made from chromosome-scale analyses of variance; e.g. using GCTA. Similarly, it would be interesting to attempt to estimate dominance and epistatic variance components. Also, the DMI model makes specific predictions about the direction of interactions. I would like to know what fraction of identified interactions fit with a model in which derived allele (inferred using an outgroup) interactions are deleterious (as opposed to alternative scenarios).

3) Although each is largely minor, there are many comments about the methodology used; justifying methods and parameter choices. I am particularly concerned about the exclusion of the X chromosome from the relatedness matrix, not least because it apparently has such a large estimated effect. This needs to be addressed.

*Reviewer #2*:

The manuscript by Turner and Harr reports the genome-wide mapping of relative testis-weight loci segregating in a natural hybrid zone of Mus m. musculus and Mus m. domesticus. They combined the high resolution Mouse Diversity Genotyping array for GWAS analysis with microarray-based gene expression analysis of the whole testis to identify the genomic loci potentially involved in reproductive isolation between both species. The logic of the experiments is difficult to follow without first reading their previous paper (Turner et al. Evolution 66:443). Even then, there are three main concerns: First and most important, the phenotype studied is complex and in respect to hybrid sterility rather vague. Testes weight (or relative TW) variation can reflect genetically determined variations without any defect, various defects in haploid phase of meiosis, intrameiotic arrests of various causes and various extent, or even a premeiotic block. If rTW is taken as the only phenotypic trait it is not surprising that the outcome is so complex and its interpretation necessarily difficult. If, at least, histological evaluation and, for example, sperm count were included, the information value would have been much higher.

Second, the use of the term “hybrid sterility loci” seems as a kind of overstatement, because none of the examined males proved to be sterile. In fact, Turner et al. 2011 Evolution paper states 'breeding data do show that hybrids have similar fertility to pure subspecies pairs' (text and Table 2). Moreover, their breeding scheme was designed to maximize the number of hybrid offspring. Thus a significant portion of variation in testes weight could reflect physiological intra- and inter-species QTLs. The argument that variation is much larger in hybrid males than in pure species is weakened by the low number of examined pure species males (See also their Evolution paper).

Third, the Dobrzhansky-Muller incompatibilities, which result in lowering testes weight can favor the involvement of such loci in reproductive isolation. The authors found 149 such chromosomal region pairs but their documentation is missing, perhaps with the exception of Figure 5, which, however, seems to show only expression DMIs (?).

In conclusion, the strength of the paper includes the first attempt to use GWAS on wild mice from a hybrid zone to infer the genetic networks involved in reproductive isolation of a young species pair. The main weak point is the inadequate phenotype selected for quantification and consequent overstatements in the interpretation of the obtained data.

Reviewer #3:

In this study, the authors use crosses among wild mice collected in the hybrid zone between M. m. musculus and M. m. domesticus to map genomic regions associated with reduced relative testis weight in hybrid males. This is a novel approach to identifying regions associated with hybrid male sterility in a system of general interest to the evolutionary genetics community. They found 26 regions across the genome, all of which interacted with at least one partner and most of which interacted with many. They also used GWAS to identify loci associated with variation in expression of testis-expressed genes. The approach seems appropriate and well executed.

My substantive comments are regarding organization, clarity and the interpretation of results. These should be remedied easily with a short round of editing.

The writing could be tighter and clearer, especially in the Abstract and Introduction. Be upfront about the advantages and disadvantages of the approach and clearly summarize the approach and major findings. Hybrid zone analyses reflect current processes in a zone of secondary contact that is relatively recent. Their significance to initial phases of reproductive isolation in allopatry is bolstered when there is overlap between this approach and QTL mapping studies. In addition, even if these specific regions were not critical in early phases, they give insight into the genetic architecture of traits that almost certainly were important.

I was often confused in the manuscript regarding which results were being referred to when e.g. which regions are being referred to in the sentence “All significant regions are involved in”?

Citations are a bit sloppy. For example, in the first paragraph of the Introduction, the authors posit that two approaches have recently substantially advanced understanding. I expected some citations of recent work and instead found only citations from Dobzhansky and Muller. Another example, the citations for “the long-recognized potential for mapping in hybrid zones...” contain only relatively recent papers. This could be easily remedied by adding “for review, see Reiseberg and Buerkle” given that that paper does review some of the older work in the system. There are many additional citations that should be considered for the statement that “islands may not always represent targets of selection...” All in all, I would suggest going through the manuscript more carefully and including citations more consistently and broadly.

More information about the system would also be useful in the Introduction, e.g. what are the three subspecies of house mice, how long have they been diverged, how old is the hybrid zone, etc. What have we learned previously? The Abstract does not actually give the names of the two subspecies.

Explain why there may and may not be overlap between loci uncovered using mapping in hybrid zones and mapping between allopatric populations.

I think this section could be re-organized to make the whole approach clearer. Explain motivation for insight into nature and timing of fertility defects. Why consider testis expression changes in the context of infertility? Give more info on what was actually done. You associated specific loci with variation in what measure of expression? Help the reader more clearly connect your results to specific insight (this refers to paragraph 1 and 2 in this section).

Candidate genes:

First sentence, identified genes from which of the previous analyses? Make the methods for identifying specific genes more explicitly methodical. First, we looked at all genes in the 26 regions with... Then we focused on the genes that were implicated in both this analysis and...

Success of GWAS:

This section seems a bit unclear. Is it GWAS that was very successful or using many different approaches? What is the success here-identification of relatively few locit or better characterization of the architecture? Be upfront about the possible downsides (environmental effects, phenotype characterization).

The entire simulation section needs more explanation and justification. Many tables and figures are devoted to this (many which could be supplemental) but the explanation is very slight. Insight into the importance of which factors in the simulations? There are three criteria for true or false positives-which ones correspond to something you considered true and which ones false? What do the results mean for the interpretation of results?

Genetic Architecture:

This section seems weak. What do you find? How does this compare with previous studies? What does this study in particular add? The reference to the Snowball effect is too slight to be effective.

---

## [Author Response]

*Analysis: Given the large effect of the X chromosome it seems critical to include X variants in the covariance matrix. We also believe that more could be made of analyses of the genetic architecture of the trait (e.g. contribution of individual chromosomes to variance in the trait using GCTA or similar software). Similarly the DMI model makes specific predictions about the direction of epistatic effects (combinations of derived alleles deleterious) which should be easy to address by polarising variants using an outgroup*.

Inclusion of X genotypes:

We repeated the mapping analysis, including X genotypes in the covariance matrix. The main effect of this change was a slightly more stringent P value cutoff to identify significant SNPs, which resulted in the identification of fewer genomic regions (26 regions excluding the X-chromosomal markers, 12 regions including the X-chromosomal markers). The positions of the regions stayed essentially the same.

Variance explained by individual chromosomes:

As suggested by reviewer 1, we used the software *gcta* to try and estimate the contribution of individual chromosomes to phenotypic variance (i.e. relative testis weight). This analysis has been pioneered on large GWAS datasets on human quantitative traits, such as height. We faced several problems with this analysis. The first problem is a conceptual one, the second a technical.

The model that *gcta* is fitting assumes that SNPs have additive effects on the trait. While this is very likely a good approximation for traits such as human height, it is not appropriate for hybrid sterility traits, because the Dobzhansky-Muller model predicts epistatic interactions between loci are necessary to observe hybrid defects. The additive genetic model would be adequate if we would map quantitative variation in testis weight within a subspecies, but not for the transgressive phenotypes in the hybrids between subspecies.

The technical problem arises when one deals with a highly structured population, such as our population from the hybrid zone. If there is an effect of population structure, SNPs on one chromosome will be correlated with the SNPs on the other chromosomes, hence estimates of variance explained individual chromosomes are overestimates. To correct for this, [89] introduced a model where the genetic relatedness matrices of all the chromosomes are fitted jointly (the –mgrm flag in gcta) to estimate the variance explained by each of the chromosomes. When we applied this version of the model, the likelihoods did not converge, even after running the maximum number of iterations allowed by the software (10,000). The most likely explanation for this problem is that our sample size is too small (185 individuals). The recommended minimum sample size for genome-partitioning analysis is 5,000 (http://gcta.freeforums.net/thread/27/partioning-autosomes).

The results presented below were obtained by running the REML procedure on each chromosome separately, incorporating the first 10 principle components (Eigenvectors) calculated over the whole autosomal dataset to correct for population structure. Similar results were obtained when Eigenvectors were calculated from the respective chromosome to correct for population structure.chromosomeV(G)/VpSEP valuechr10.5413090.1156481.66E-05**chr2****0.698186****0.09151****3.09E-06**chr30.3710460.1399060.006895chr40.5786850.1134676.23E-05chr50.2797610.1435280.02633chr60.5466150.1095883.64E-07chr70.5477680.105442.16E-06chr80.3473580.1338620.001698chr90.4168490.1433850.005933chr100.3112710.1646130.05873chr110.4577160.1312590.002731chr120.4465140.117824.18E-07chr130.3062720.1341670.004079chr140.5211040.1166298.90E-08chr150.3575160.1218790.0002658chr160.543880.1122186.10E-06**chr17****0.644497****0.091807****8.874E-09**chr180.2089510.1459250.1124chr190.0567780.1094880.2996**chrX****0.752231****0.071999****9.09E-10**

The heritability estimates (i.e. the proportion of phenotypic variation due to additive genetic factors, V(G)/Vp) are not reliable due to the complications stated above. However, in both sets of analyses, chromosomes 2, 17 and X explained most of the variation. These results are consistent with our mapping analysis, which identified significant GWAS regions located on those chromosomes.

Ancestral vs. derived sterility alleles:

We agree that determining if sterility alleles are ancestral or derived would be of great interest. However, significant SNPs identified by GWAS are unlikely to include the causal mutations for mapped phenotypes. Without knowing the causal variant, it is not straightforward to categorize sterility alleles as ancestral or derived. Nevertheless, we compared genotypes in significant regions from *M. m. musculus* and *M. m. domesticus* populations to *M. spretus, M. spicilegus and M. mattheyii.* Each region comprises numerous SNPs that can be polarized and additional sites with shared polymorphisms between *musculus* and *domesticus*. Thus, it will not be possible to determine which alleles are ancestral vs. derived until future studies identify causal genes/mutations.

*Interpretation: The phenotype studied is not hybrid sterility (despite the Title). There is actually no direct evidence for an association between the trait studied and fitness. Indeed the statement that there were no significant results for sperm count is a little worrying. However, there is limited evidence that the phenotypes of the hybrids are typically outside the range of normal variation within the species. It is important to note these caveats*.

We agree that that there was a need for additional explanation of links between the mapped phenotype and hybrid sterility. We discuss this issue in detail now and note caveats in the “phenotyping” section of Methods, and the “sterility-associated phenotypes” and “effect size” sections of Results. In addition, we mapped another sterility-associated phenotype: testis expression PC1.

*Presentation: Without fine-mapping or replication, the ability to identify/localise specific genes as being important in the trait is limited. For this reason, we feel the emphasis on long lists of rather weakly-supported candidate genes is misplaced. There are also issues with formatting, clarity of figures and citations*.

We agree. We simplified annotation of the GWAS regions, reporting only genes with strong evidence for roles in reproduction (Tables 1 and 2) and removed most of the discussion of candidate genes from the text.

We made several major changes to the organization of the manuscript to improve clarity. First, we edited the Introduction, incorporating suggestions from reviewers to better explain the motivation and logic of the study. Second, we separated Results and Discussion and added more subheadings. Third, we reduced the number of figures in the main text to four.

Additional changes:

We made several changes in response to concerns of individual reviewers, notable examples include:

Significance thresholds: We clarified the motivation for reporting results using a permissive (FDR <0.1) threshold and using multiple lines of evidence to identify loci likely to be true positives. We report estimates of the false positive rate using stringent and permissive thresholds from simulations.

Tests for interactions:

Instead of testing for interactions with MCMCglmm, we used a mixed model approach similar to the GEMMA framework used to identify single-SNP associations.

Overlap between GWAS regions and sterility QTL:

We performed permutation tests to determine if associations between GWAS regions and previously reported candidate sterility regions were non-random. It was not possible to test for non-random concordance between genetic interactions identified in this study and those reported in F2 hybrids by permutation because individual GWAS regions have multiple partners and the number of possible pairwise interactions between SNPs varies widely across region pairs. In the absence of statistical support for the overlapping pattern, we decreased emphasis on this result by including the figure showing overlap as a supplemental figure.

Yang, J., et al. 2011. Genome partitioning of genetic variation for complex traits using common SNPs. Nat. Genet. 43:519–525.